# Particle fluxes by subtropical pelagic communities under ocean alkalinity enhancement

Authors: Philipp Suessle[1], Jan Taucher[1], Silvan Urs Goldenberg[1], Moritz Baumann[1], Kristian Spilling[2,3], Andrea Noche-Ferreira[3,4], Mari Vanharanta[2,5], Ulf Riebesell[1]

Affiliations

[1] GEOMAR Helmholtz Centre for Ocean Research Kiel, Wischhofstrasse 1-3, 24148 Kiel, Germany

[2] Marine and Freshwater Solutions, Finnish Environment Institute, Helsinki, Finland

[3] Centre for Coastal Research, University of Agder, Kristiansand, Norway

[4] Department of Natural Sciences, University of Agder, Kristiansand, Norway

[5] Tvärminne Zoological Station, University of Helsinki, Hanko, Finland

Corresponding author: Philipp Suessle (philippsuessle@gmx.de)

**Abstract**

Ocean alkalinity enhancement (OAE) has been proposed as a carbon dioxide removal technology (CDR) allowing for long term storage of carbon dioxide in the ocean. By changing the carbonate speciation in seawater, OAE may potentially alter marine ecosystems with implications for the biological carbon pump. Using mesocosms in the subtropical North Atlantic, we provide first empirical insights into impacts of carbonate-based OAE on the vertical flux and attenuation of sinking particles in an oligotrophic plankton community. We enhanced total alkalinity (TA) in increments of 300 µmol kg$^{-1}$, reaching up to $\Delta TA$ = 2400 µmol kg$^{-1}$ compared to ambient TA. We applied a $p$CO$_2$-equilibrated OAE approach, i.e. dissolved inorganic carbon (DIC) was raised simultaneously with TA to maintain seawater $p$CO$_2$ in equilibrium with the atmosphere, thereby keeping perturbations of seawater carbonate chemistry moderate. The vertical flux of major elements including carbon, nitrogen, phosphorus and silicon, as well as their stoichiometric ratios (e.g. carbon-to-nitrogen) remained unaffected over 29 days of OAE. The particle properties controlling the flux attenuation including sinking velocities and remineralization rates also remained unaffected by OAE. However, we observed abiotic mineral precipitation at high OAE levels ($\Delta TA$ = 1800 µmol kg$^{-1}$ and higher) that resulted in a substantial increase in PIC formation. The associated consumption of alkalinity reduces the efficiency of CO$_2$ removal and emphasizes the importance of maintaining OAE within a carefully defined operating range. Our findings suggest that carbon export by oligotrophic plankton communities is insensitive to OAE perturbations using a CO$_2$ pre-equilibrated approach. The integrity of ecosystem services is a prerequisite for large-scale application and should be further tested across a variety of nutrient-regimes and for less idealized OAE approaches.

**Keywords:** ocean alkalinity enhancement, carbon export flux, biological carbon pump, export flux stoichiometry, remineralization, sinking velocities, climate change, nature-based solution

# 1 Introduction

Carbon dioxide removal (CDR) on a grand scale is likely to be needed to restrict global warming to between 1.5 to 2°C (IPCC, 2021). Oceans play a key role in buffering the climate by absorbing heat and $CO_2$ and ocean-based CDR could further enlarge the oceanic carbon pool (GESAMP, 2019). Ocean alkalinity enhancement (OAE) has been proposed as one such measure (NAS, 2021). Simulating natural rock weathering, it elevates the oceans' total alkalinity (TA) by dissolving large amounts of minerals in seawater (Kheshgi, 1995; González and Ilyina, 2016). The associated change in carbonate speciation increases the capacity of seawater to take up $CO_2$ (Köhler et al., 2010; Hartmann et al., 2013). OAE has several potential benefits compared to other CDR technologies (Taylor et al., 2016). It does not compete for freshwater-, land-, or nutrient-use (Smith et al., 2016), can rely on existing infrastructure from the mining industry (Renforth and Henderson, 2017), is applicable to large regions of the open ocean (Köhler et al., 2010) and additionally counters effects of ocean acidification (Doney et al., 2009; Gattuso et al., 2015). Different modelling studies suggest a potential $CO_2$ uptake by OAE ranging from $14 - 41$ Gt $CO_2$ year$^{-1}$ (Paquay and Zeebe, 2013; González and Ilyina, 2016; Feng et al., 2017). For comparison, natural rock weathering currently locks up 1.1 Gt $CO_2$ year$^{-1}$ (Ciais et al., 2014).

Before large scale deployment, the safety of OAE for marine organisms and ecosystem services has to be assured. OAE-related changes in pH, ion concentrations and the carbonate saturation state ($\Omega$) would be permanent and may be elevated to well above current and pre-industrial levels (Feng et al., 2017; Renforth and Henderson, 2017). Such shifts in carbonate speciation might influence marine organisms (Bach et al., 2019a), analogous to effects observed for ocean acidification. In turn, calcifying organisms may theoretically benefit from e.g. lime-based alkalinity enhancement, as the enhanced $CaCO_3$ saturation state (through excess of $Ca^{2+}$ and $CO_3^{2-}$ ions) and increased pH may facilitate $CaCO_3$ precipitation, allowing calcifiers to allocate more energy towards growth or other metabolic processes (Jokiel, 2011; Monteiro et al., 2016; Bach, 2015). Riebesell et al. (2017) demonstrated that varying pH levels on natural communities caused large differences in population sizes of the prominent calcifying phytoplankton species *Emiliania huxleyi*. If there are ecological impacts of OAE, these would likely propagate through communities altering ecosystem services such as the biological carbon pump, which sequesters $CO_2$ into the deep ocean for hundreds to thousands of years (Boyd et al., 1999).

Several studies have revealed the influence of abiotic conditions on the magnitude and efficiency with which particulate organic carbon (POC) from photosynthetic biomass production is exported by gravitational settling (Schmittner et al., 2008; Taucher and Oschlies, 2011; Bopp et al., 2013; Martínez-García et al., 2014). Assuming a proliferation of calcifiers, organic carbon export could increase in response to higher carbonate ballasting (Honjo et al., 2008). Carbonate ballast is generally described by the fraction of particulate inorganic carbon (PIC) to particulate organic carbon (POC) (Klaas and Archer, 2002).The mineral's excess density (Francois et al., 2002) would increase sinking velocities (SV) of the export flux (Bach et al., 2016). With more carbonate ballasting, particles would on average sink deeper, thus storing atmospheric $CO_2$ for longer (Kwon et al., 2009). Additionally, particles containing $CaCO_3$ have shown reduced carbon remineralization rates ($C_{remin}$), possibly due to protection of POC in mineral shells, further increasing carbon transport to depth (Engel et al., 2009). A

proliferation of calcifiers, however could also hamper OAE's capability to store atmospheric carbon (Bach et al., 2019a). The $CO_2$ release associated with calcification is known as the carbonate counter pump and reflects a positive feedback on atmospheric carbon levels (Koeve, 2002; Salter et al., 2014). Recent laboratory-based OAE experiments (Moras et al., 2021; Hartmann et al., 2023) demonstrated an alkalinity consumption due to abiotic PIC precipitation, further highlighting its importance for larger scale OAE applications and implications on carbon export.

The $CO_2$ sequestration potential of the biological pump also hinges on the stoichiometric ratios of carbon to other macronutrients within the export flux (Shaffer, 1996; Schneider et al., 2003; Passow and Carlson, 2012). An OAE induced community restructuring could change the nitrogen-need per unit of exported carbon, binding the potential magnitude of carbon export to the supply of the limiting nutrient (Hessen et al., 2004). Such a change in carbon to particulate organic nitrogen ratios (POC:PON) has also been noted in response to changing heterotrophic processes under ocean acidification (Taucher et

al., 2021), implying that similar effects might occur under OAE and further highlights the importance of stoichiometric ratios for carbon sequestration. Whether or not OAE induces substantial changes in species communities and their mediation of the oceans natural carbon sequestration via the biological pump remains to be seen.

    Theoretically, OAE is conceivable in two different approaches. While one merely enhances alkalinity, letting $CO_2$ equilibrate with the atmosphere post application, the other approach entails adding solutions where $CO_2$ is already at equilibrium with the

atmosphere, resulting in milder carbonate chemistry perturbations (so called pre-equilibrated approach). Here, we examined influences of $CO_2$ pre-equilibrated, carbonate-based ocean alkalinity enhancement on drivers of the biological carbon pump. Using mesocosms in the subtropical North Atlantic, we examined whether OAE (with a doubling of the natural seawaters alkalinity) changes the magnitude and the stoichiometry of the carbon export flux, alters particle properties and if so, to what extent alkalinity consumption or enhanced ballasting through calcification occurs. With this first study on carbon export

properties in a natural plankton assemblage, we provide urgently needed understanding on the biogeochemical effects of OAE.

## 2 Material and Methods

### 2.1 Experimental setup

On the 8th of September 2021, nine in situ mesocosms were deployed in the harbor of Taliarte (27° 59′ 24″ N, 15° 22′ 8″ W; east coast Gran Canaria, Spain). Each mesocosm was comprised of a 2.5 m long transparent cylindrical polyurethane foil (Ø = 2 m) with a 1.5 m long conical sediment trap attached below. A polyethylene frame kept the mesocosm bags afloat and between sampling, mooring and pulley systems allowed to move the construction 2 m away from the pier, thereby reducing its influence on the light regime. The mesocosms were filled with seawater from ~100 m outside the harbor (water depth at collection site ~ 20 m). Using a peristaltic pump, water from 2 – 10 m depth (within the surface mixed layer, Barton et al., 1998, but avoiding benthic impact) was filtered (< 3 mm) and gathered in an intermediate storage volume, from where each mesocosm simultaneously received ~8 m$^3$ in total. In order to reduce stress on the organisms, the speed of the peristaltic pump averaged ~17.35 L min$^{-1}$ and the entire process lasted ~7 h (for further technical details see Bach et al. (2019b)). Filling was done on the 10th of September and marks Day 0 of the experiment. CTD casts (CTD60M, Sea & Sun Technology GmbH, Trappenkamp, Germany) for temperature, salinity, pH, turbidity, oxygen ($O_2$) and photosynthetically active radiation prior to treatment addition assured, that the environmental conditions of water pumped into the mesocosms were as close as possible to those expected for the surface mixed layer of the given oceanographic setting and season (Barton et al., 1998). OAE was simulated in a $pCO_2$-equilibrated OAE approach, meaning that dissolved inorganic carbon (DIC) was raised simultaneously with TA to maintain seawater $pCO_2$ in equilibrium with the atmosphere, thereby keeping perturbations of seawater carbonate chemistry (e.g. pH, $\Omega_{Ar}$) as low as possible. This contrasts non-equilibrated OAE, in which only TA would be increased (while DIC remains constant), leading to much larger transient changes in carbonate chemistry until seawater $pCO_2$ reaches equilibrium with the atmosphere due to air-sea gas exchange. Accordingly, a pre-equilibration of OAE solutions with $CO_2$ represent a least impactful, most optimistic OAE scenario. The OAE treatment in our mesocosm experiment was realized by adding $CO_2$-equilibrated solutions of sodium-carbonate and –bicarbonate ($NaHCO_3$/$Na_2CO_3$ in deionized water) on Day 4. With a custom-built device (see Riebesell et al., 2013 for technical details) each 22 L solution was spread evenly across the water column within 2.5 min. While one mesocosm was only treated with deionized water (control), the remaining eight received increments of $\Delta TA = 300$ µmol kg$^{-1}$, resulting in 4689.3 µmol kg$^{-1}$ (Day 7) for the highest treatment level. This equaled double the alkalinity of the natural seawater. For details on changes in carbonate chemistry parameters see Table S1. The mooring of the mesocosms was randomized along the pier of the harbor and the experimental setup enabled us to monitor pelagic communities and biogeochemical responses to OAE for 29 days.

**Table 1 |** Experimental design displaying the increments of TA increase relative to the natural background alkalinity of 2402.7 µmol kg$^{-1}$

| OAE ($\Delta TA$ µmol kg$^{-1}$) | 0 | 300 | 600 | 900 | 1200 | 1500 | 1800 | 2100 | 2400 |
|---|---|---|---|---|---|---|---|---|---|
| Symbol | ▲ | ● | ■ | ◆ | ▼ | ● | ■ | ◆ | ▲ |

## 2.2 Sampling procedure and maintenance

In general, sampling was done every 2 days between 09:00 and 12:00 a.m. Additionally, to follow any biological responses related to the filling process, daily sampling was conducted prior to the alkalinity addition. Bulk water samples were taken with a polypropylene pipe, integrating the upper 2.3 m of the water column. On each sampling occasion, 40 – 60 L (8 – 12 tubes) of water were sampled per mesocosm. The water was stored dark and at in-situ temperature in several carboys and transported back to the lab-facilities for analysis of chlorophyll $a$ (Chl $a$), TA and dissolved inorganic carbon (DIC), dissolved inorganic nutrients ($NO_3^- + NO_2^-$, $PO_4^{3-}$, $Si(OH)_4$), as well as water column particulate matter ($PM_{WC}$). Other measurements conducted during the study included analysis of primary productivity, photosynthetic pigment analysis, phyto- and microzooplankton analysis and prokaryotic heterotrophic production. A CTD60M (Sea & Sun Technology GmbH, Trappenkamp, Germany) was cast for additional abiotic parameters such as temperature, salinity, pH, turbidity, oxygen ($O_2$) and photosynthetically active radiation. Particulate matter from the sediment trap ($PM_{ST}$) was collected using a manual vacuum pump, not exceeding 0.3 bar during the pumping. Stored in 5 L glass bottles (Schott Scandinavia A/S, Kgs. Lyngby, Denmark), the sediment was kept dark until further processing in the lab (see Boxhammer et al. (2016) for technical details). Homogenous subsamples were taken after resuspension of the sediment by gentle rotation of the glass bottles. The subsamples were used for determination of carbon-specific remineralization rates ($C_{remin}$) and particle sinking velocities (SV) as described below.

The inside of the mesocosm walls was cleaned every 4 to 6 days with brushes to prevent nutrient consumption and shading from organisms growing on the walls (for a detailed experimental schedule see Paul et al., 2024). To further counteract shading, cleaning from the outside was done three times during the experiment by divers equipped with brushes. During cleaning on Day 16, the sediment hose of mesocosm $\Delta TA_{900}$ detached. Pumping $PM_{ST}$ on the subsequent sampling day was not possible, thus SV measurements on Day 17 are missing for this mesocosm. Measurements of SV and $C_{remin}$ on Day 19 however are not discarded, as they did not significantly differ from other treatment levels on that day. Yet, interpretation should be done with care, as the period for $PM_{ST}$ settlement was ~5 h less than usual.

## 2.3 Drivers of the biological carbon pump

### 2.3.1 Export fluxes and stoichiometry

$PM_{ST}$ material was prepared for analysis of total particulate carbon and nitrogen ($TPC/N_{ST}$), particulate organic carbon, nitrogen and phosphorus ($POC/N/P_{ST}$) as well as biogenic silica ($BSi_{ST}$) after subsampling. First of all, particles were separated from the seawater. To increase flocculation and coagulation, 3 mol $L^{-1}$ ferric chloride ($FeCl_3$) were added to the 5 L bottles with sediment material, followed by the addition of 3 mol $L^{-1}$ NaOH to counteract the decreasing pH, promoting an effective particle collection process as shown by previous research (Boxhammer et al., 2016). After 1 h of settlement, the supernatant was decanted and the remaining flocculated material was centrifuged for 10 min at 5.200 $g$ in a 6–16KS centrifuge (Sigma Laborzentrifugen GmbH, Osterode am Harz, Germany). A second centrifugation step for 10 min at 5.000 $g$ in a 3K12 centrifuge (Sigma) yielded solid sediment pellets. The pellets were stored at -20°C and transported back to Kiel for further

processing. At GEOMAR, samples were freeze-dried to remove remaining moisture and ground to a fine homogenous powder in a cell mill (Edmund Bühler GmbH, Bodelshausen, Germany). The sediment powder was stored dark and cool in glass or plastic vials until further processing. Sediment powder was transferred into silver or respectively tin capsules, and either acidified with 1 mol L$^{-1}$ HCl and dried at 50°C over-night in order to measure POC/N$_{ST}$, or proceeded unacidified for TPC/N$_{ST}$ analysis. Duplicates were measured on a CN analyzer (Euro EA-CN, HEKAtech GmbH, Wegberg, Germany) according to (Sharp, 1974). PIC$_{ST}$ concentrations were calculated from the difference of TPC$_{ST}$ to POC$_{ST}$. BSi$_{ST}$ and POP$_{ST}$ concentrations were measured spectrophotometrically according to Hansen and Koroleff (1999), after subsampling ~ 2 mg of the sediment. Prior to that, POP samples were pressure-cooked in 40 ml of deionized water including an oxidizing agent (Oxisolv, Merck).

### 2.3.2 Particle sinking velocities

Sinking velocities (SV) of sediment trap material were determined every two days by means of video-microscopy. A repurposed FlowCam (Fluid Imaging Technologies Inc., Scarborough, United States) was used to capture images of particles during gravitational sinking. A vertically mounted sinking chamber (cuvette with the dimensions 10 x 10 x 350 mm) was placed in front of the camera (see Bach et al. (2012) for technical details). Sediment subsamples were inspected visually and diluted (with filtered seawater < 0.2 μm) in a small cup according to their particle density (1:1 – 1:40). Diluted samples were loaded into the sinking chamber by a pipette and pileus ball attached atop. Fluid advection within the sinking chamber could be ruled out by ensuring air-tightness. Measurements were carried out at in situ temperatures (~22°C) and additional ventilation minimized convection within the sinking chamber. Particles between 25 – 1000 μm were identified with a 2x magnification and 20 frames-per-second, and their sinking velocities were calculated by applying a linear regression to the vertical position against time. Several optical parameters allowed to identify the multiple frames of a single particle during gravitational sinking and calculations in the MATLAB script (version R2021b; provided by Bach et al. (2012), see "Evaluation of sinking velocities") were adjusted to the more recent version of the FlowCam 8000. These were corrected for temperature differences during measurements and for wall effects of the sinking chamber following Ristow (1997). Partially captured particles, or those out of focus were excluded within the MATLAB script, prior to the SV calculation. Measurements were run for 20 min and subsequent data analysis was carried out with the programming software R (R Core Team, 2021) using the package *tidyverse* (Wickham et al., 2019) in RStudio (Version 4.1.1.). It is important to note that measured particle size spectra are derived from ex situ measurements and thus do not accurately represent the in situ size distribution. Nevertheless, they allowed to follow in situ particle dynamics due to e.g., aggregation, packing or grazing closely.

### 2.3.3 Remineralization rates of sinking particles

Carbon specific remineralization rates were determined every 4 – 6 days. During the water column sampling, four replicate and three control bottles for each mesocosm were filled headspace-free. Upon arrival in the temperature-controlled lab (in situ temperature of the respective day), the respiration bottles were acclimatized in a water bath for 1 h. Depending on particle density, 0.5 – 10 ml of homogenized sediment subsample of the respective mesocosm were slowly injected into four replicated

bottles. Control bottles (n = 3) did not receive any sediments. Subsequently, all bottles were dark-incubated on a plankton wheel (~1 rpm) for 22 – 26 h. Non-invasive $O_2$ measurements were carried out in regular intervals (appr. every 4 h) using a fiber optical measurement device (Fibox4 Trace, PreSens – Precision Sensing GmbH, Regensburg, Germany) and PSt3 sensor spots (PreSens) mounted to the inside walls of the incubated bottles. Corrections for atmospheric pressure and temperature (using a dummy bottle) were automatically done by the Fibox4. The Fibox4 salinity correction was set to the mean mesocosm salinity of the respective day (CTD measurement).

Incubations were terminated with the collection of POC (sediment + suspended) within the bottles on pre-combusted glass fiber filters (0.7 μm, Whatman). The carbon content analysis was carried out as described for $POC_{WC}$ (see Sect. 2.3.4). The carbon specific remineralization rate of sedimented particulate matter ($C_{remin}$ in $d^{-1}$) was calculated as follows:

$$C_{remin} = \frac{(r * RQ)}{(POC + r * RQ * \Delta t)} \tag{1}$$

The $O_2$ consumption rate (r in $\mu mol\ O_2\ L^{-1}\ d^{-1}$) was normalized to the POC ($\mu mol\ C\ L^{-1}$) contained within the bottles, which was corrected for the duration of the incubation ($\Delta t$ in d). The respiration quotient (RQ in $mol\ C\ mol\ O_2^{-1}$) equalled 1, since the consumption of $O_2$ is commonly assumed to be equal to the production of $CO_2$ during organic matter turnover (Ploug and Grossart, 2000; Iversen and Ploug, 2013; Bach et al., 2019c). To differentiate the influence of sediment respiration from ambient seawater $O_2$ consumption, $C_{remin}$ was corrected for the mean blank bottle values.

### 2.3.4 Phytoplankton biomass, abundances and water column stoichiometry

Subsampling for $TPC/N_{WC}$, $POC/N_{WC}$ and Chl $a$ was achieved by filtering the water column samples onto separate pre-combusted glass fiber filters (0.7 μm, Whatman). To remove inorganic carbon, $POC/N_{WC}$ filters were acidified for ~30 sec. using 1 mol $L^{-1}$ HCl. Additionally, unacidified filters for $TPC/N_{WC}$ were collected. Both filter types were stored in pre-combusted glass petri-dishes and dried at 60°C overnight. All filters were packed into tin-cups (8 x 8 x 15 mm, LabNeed GmbH, Nidderau, Germany), stored in a desiccator and transported back to Kiel. Carbon content of filters was determined with a CN analyzer (Euro EA-CN, HEKAtech) as described for the sediment trap samples. $PIC_{WC}$ concentrations were calculated as described for the sediment trap samples.

Chl $a$ was extracted in plastic vials using acetone (90%) and subsequent homogenization in a cell mill with glass beads (5 min; Vibrogen, Berlin, Germany). The suspension was centrifuged (10 min, 800 $g$, 4°C; Universal 320, Hettich, Bäch, Germany) and the resulting supernatant was analyzed with a fluorometer (TURNER Trilogy, Turnerdesigns, San Jose, United States) according to Welschmeyer (1994). Spinach Chl $a$ extract (Sigma-Aldrich, St. Louis, United States) was used to calibrate the fluorometer (Strickland and Parsons, 1972).

Phyto- and microzooplankton abundances were determined by microscope from bulk water samples according to Utermöhl (1958) after fixating in basics Lugols solution and letting samples settle in a chamber for 24 h.

### 2.3.5 Carbonate chemistry and nutrient concentrations

TA and DIC samples were taken with an overflow of ~2 times the final sampling volume, carefully avoiding air-inclusion.
Within 12 h, samples were measured at room temperature. To remove $PIC_{WC}$, both TA and DIC samples were filtered (0.2 μm, Whatman). TA was determined in duplicates with hydrochloric acid (0.05 M) in a two-stage open cell potentiometric titrator (Compact Titrosampler 862, Metrohm, Munich, Germany).

DIC was determined in triplicates by infrared absorption using an AIRICA system (MARIANDA, Kiel, Germany) with a differential gas analyzer (LI-7000, LI-COR Biosciences GmbH, Bad Homburg, Germany). Values of DIC and TA were calibrated against certified reference material (batch no.: 143, 190; Dickson, 2010). Further carbonate chemistry parameters were calculated with the excel macro "*CO2SYS*" (Pierrot et al., 2011) using the (water column averaged) temperature and salinity CTD measurements for correction. The carbonate dissociation constants ($K_1$ and $K_2$) from Lueker et al. (2000) were chosen.

Dissolved inorganic nutrient concentrations ($NO_3^- + NO_2^-$, $PO_4^{3-}$, $Si(OH)_4$), were measured spectrophotometrically on a five channel continuous flow analyzer (QuAAtro AutoAnalyzer, SEAL Analytical Inc., Mequon, United States) with a fluorescence detector (FP-2020, JASCO, Pfungstadt, Germany). Approaches of Murphy and Riley (1962), Mullin and Riley (1955) and Morris and Riley (1963) were followed for all colorimetric methods and refractive index correction was done according to Coverly et al. (2012).

### 2.3.6 Statistical analysis

Linear mixed effects models were used to assess the OAE effect on the different export-related response variables. OAE was set as a continuous fixed effect, Day as categorical fixed effect and Mesocosm as random effect (*nlme* package; Pinheiro et al., 2022). The interaction term of OAE × Day was set as an additional fixed effect to test how the OAE effect changed over time. For all analyses, assumptions were checked using residual-vs-fitted plots (homogeneity of variances and QQ Plots (normality of residuals; *performance* package; Lüdecke et al., 2021). Their violation was corrected for with data transformations (Table S3). All analyses were conducted at a significance level of $\alpha = 0.05$ using R (R Core Team, 2021). Additionally, averages over experimental phases were calculated for specific response variables and their dependency on OAE tested using simple linear regressions. The highest OAE level ($\Delta TA_{2400}$) was excluded from all statistical analyses as it developed considerable abiotic precipitation which interfered with the interpretation of the OAEs effect on the biological pump. A *Welch´s t-test* on TA between two phases of the experiment (Day 4 – 17 vs. Day 19 – 33, Table S2) confirmed its standing as an extreme outlier.

## 3 Results

### 3.1 Achieved alkalinity enhancement and plankton community biomass

We successfully applied OAE to the mesocosms and target alkalinity levels remained stable throughout the experiment (Fig. 1a). An exception was the highest OAE level, which only remained stable until Day 17, after which $\Delta TA_{2400}$ continuously declined, dropping below $\Delta TA_{2100}$ on Day 33. This alkalinity consumption was induced by abiotic calcium carbonate precipitation (see Sect. 4.2). Phytoplankton biomass approximated by chlorophyll $a$ and particulate organic carbon in the water column ($POC_{WC}$) did not change with respect to OAE (Fig. 1b,c, Table S4). It remained low in the first half of the experiment (Day 3 – 21 Fig. S3), where after chlorophyll $a$ and $POC_{WC}$ concentrations unexpectedly increased in some mesocosms ($\Delta TA_{600}$, $\Delta TA_{900}$, $\Delta TA_{1500}$, and $\Delta TA_{1800}$). These blooms occurred despite oligotrophic conditions throughout the entire experiment (Fig. S2) and were likely caused by a mixotrophic haptophyte species of *Chrysochromulina spp.* (Xin et al., 2025).

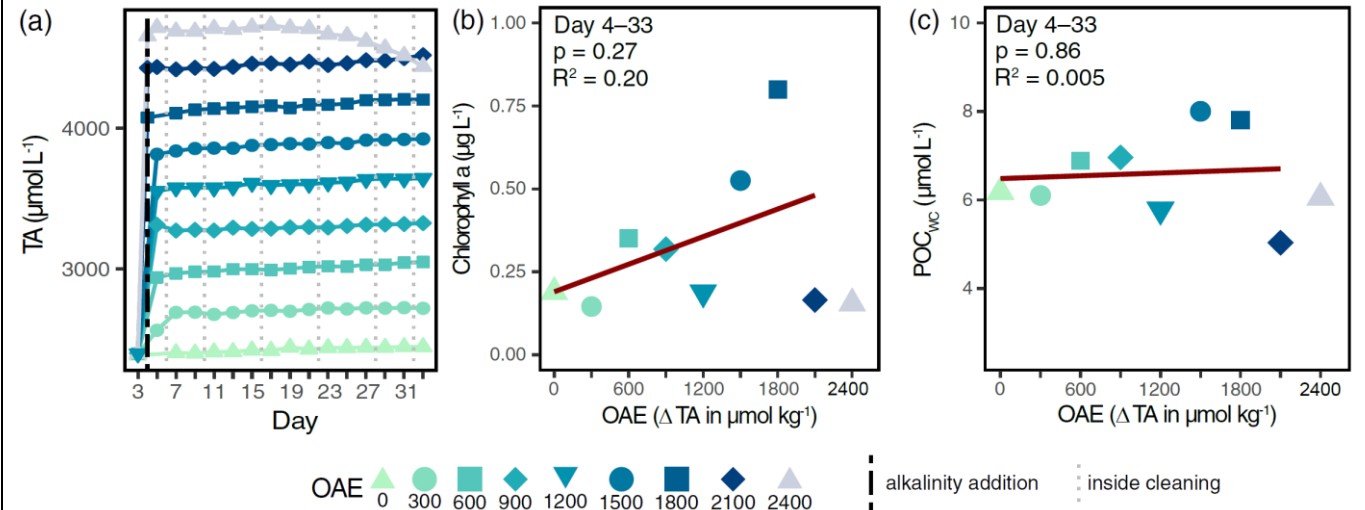

**Figure 1| Abiotic and biotic responses of the water column to the application of OAE.** Achieved alkalinity enhancement and its stability over time (a). The vertical black line indicates the alkalinity addition and the grey lines the inside cleaning of the mesocosm walls. Chlorophyll $a$ (b) and carbon biomass (c) of the plankton community. Averages over the treatment period are employed in linear regressions.

## 3.2 Organic matter fluxes and stoichiometry

Ocean alkalinity enhancement did not influence the magnitude of the particulate organic carbon flux ($POC_{ST}$), neither the cumulated $POC_{ST}$ over the entire experiment, nor on a daily basis (Table S3a). Irrespective of OAE, carbon export fluxes decreased over time (Fig. 2a). Higher initial fluxes (Day 7 & 9) reflected the decreasing phytoplankton biomass of preceding days and thereupon, an average 8-fold decrease was visible until day 27. Other major elemental fluxes ($PON_{ST}$, $POP_{ST}$ and $BSi_{ST}$) were likewise not influenced by OAE (Fig. 2c,e,f, Table S3i,j,k) and had similar temporal trends as POC fluxes (Fig. S4b,c,d). Whether the export of the major elements was affected within the blooming mesocosms ($\Delta TA_{600}$, $\Delta TA_{900}$, $\Delta TA_{1500}$, and $\Delta TA_{1800}$, Fig. S4) remains unclear, yet the influence of wall cleaning is apparent from Day 17 onwards, as wall growth intensified. Carbon-to-nitrogen ratios ($POC_{ST}$:$PON_{ST}$) of exported matter did not change with respect to OAE (Fig. 2d). The stoichiometric composition stayed constant throughout the experiment (Table S3b) and generally well above the canonical Redfield ratio of 6.6 (Fig. S4a). Likewise, opal ballasting ($BSi_{ST}$:$POC_{ST}$) was unaffected by OAE (Fig. 2g, Fig. S4e, Table S3l).

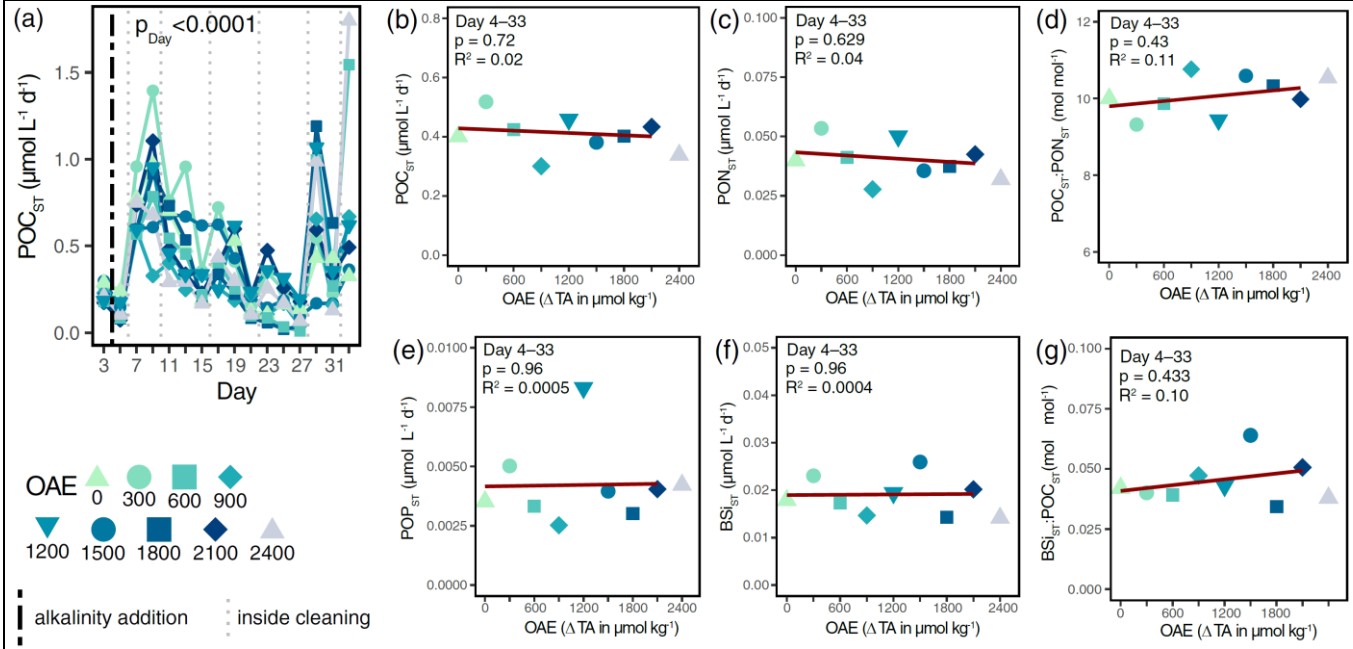

**Figure 2| Organic matter fluxes and stoichiometry under the application of OAE.** Organic carbon flux and its response to OAE over time (a) and averaged across the treatment period (b). The p-value in (a) represents the output of linear mixed effects model (Table S3a). The vertical black line indicates the alkalinity addition and the grey lines the inside cleaning of the mesocosm walls. Other major elemental fluxes ($PON_{ST}$, $POP_{ST}$, $BSi_{ST}$) in response to OAE averaged over the treatment period (c,e,f). Quality and ballasting of the export flux ($POC_{ST}$:$PON_{ST}$, $BSi_{ST}$:$POC_{ST}$) averaged over the treatment period employed as linear regressions (d, g).

## 3.3 Particle sinking velocities and remineralization rates

OAE had no effect on either particle sinking velocities or remineralization rates of sinking material, leaving the potential flux attenuation with depth unchanged (Fig. 3a, c and Table S3f,m). Particle sinking velocities (SV) were generally low (10 – 35 m d$^{-1}$) and did not respond to OAE, neither overall (Fig. 3a), nor for specific particle size classes (Fig. S5). Sinking velocities of $\Delta TA_{2400}$ exceeded those of all other alkalinity levels from Day 15 onwards and are expected to be caused by increased ballasting due to the abiotically precipitated carbonate mineral (see Sect. 4.2). Remineralization rates ($C_{remin}$) did not change under OAE (Fig. 3c, Table S3f), staying relatively constant with carbon turnover varying between 7 and 18% per day (Fig. 3c). Noticeably higher remineralization rates of $\Delta TA_{2400}$ on Day 27 and 33 were not paralleled by higher $O_2$ consumption rates across incubations (Fig. S7) and were probably caused by an unresolved sample processing error due to the abiotic precipitation (see Sect. 4.2).

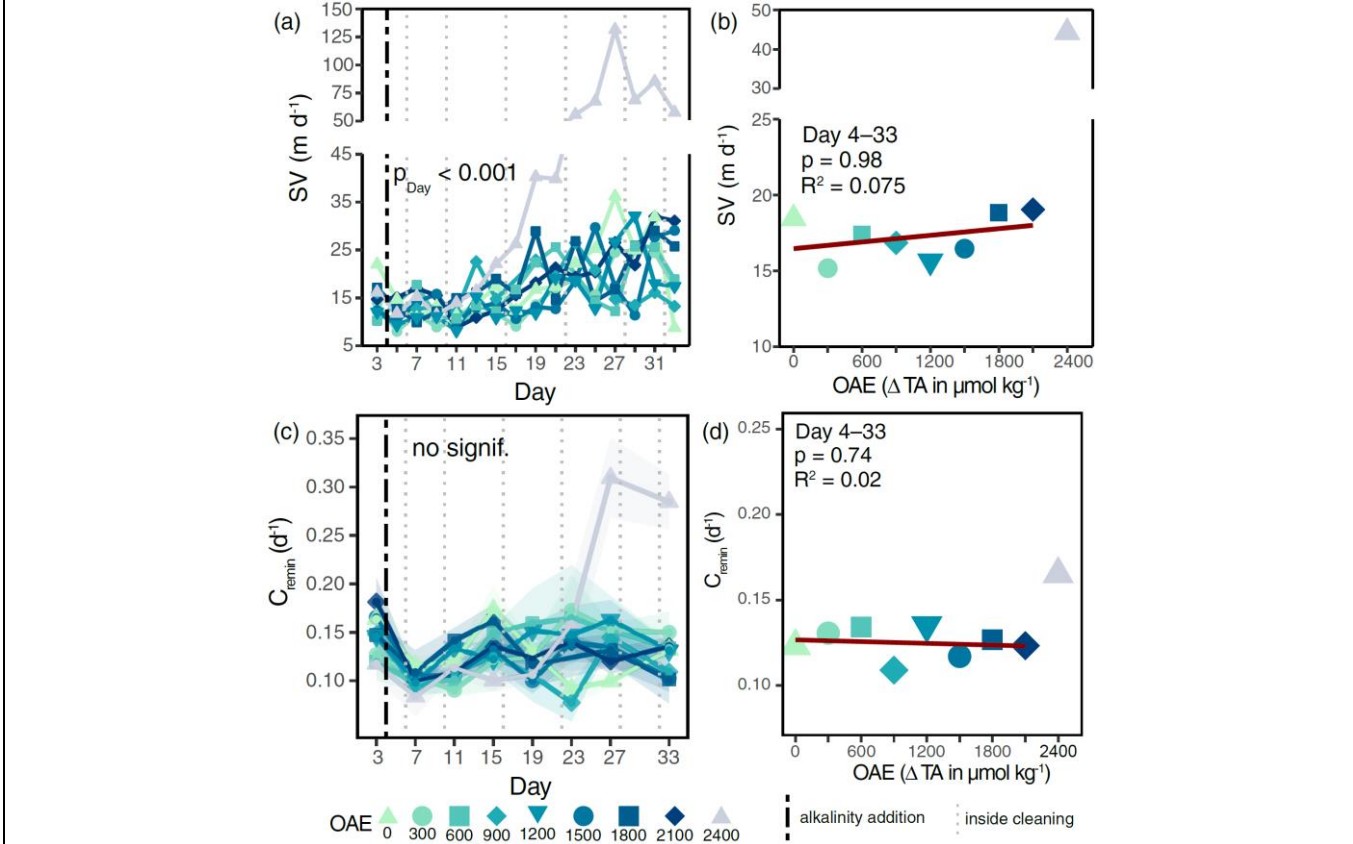

**Figure 3| Depth attenuation of the organic matter flux under the application of OAE.** Development of average particle sinking velocities under OAE over time (a) and as averages over the treatment period (b). Carbon specific remineralization rates (c) and averages under OAE (d). Shaded areas in (c) indicate standard deviations of measurements ($n_{min}$ = 3). p-values in (a) and (c) represent the output of linear mixed effects models (Table S3f,m). The vertical black line indicates the alkalinity addition and the grey lines the inside cleaning of the mesocosm walls (a,c).

## 3.4 Abiotic precipitation, inorganic carbon fluxes and carbonate ballasting

Particulate inorganic carbon export ($PIC_{ST}$) and carbonate ballasting ($PIC_{ST}:POC_{ST}$) increased under OAE (Fig. 4a, Fig. S6a, Table S3g,h). Between Days 19 and 31, average $PIC_{ST}$ fluxes almost tripled from ambient seawater alkalinity to $\Delta TA_{2100}$ (Fig. 4b), while carbonate ballasting more than doubled (Fig. S6b). Additionally, the $PIC_{ST}$ flux in $\Delta TA_{2400}$ exceeded those of all other alkalinity levels from Day 17 onwards and $pCO_2$ was highest (e.g., 541 µatm on Day 33). High $PIC_{ST}$ fluxes coincided with a white precipitate (Fig. S8d), especially apparent on the walls of the mesocosm $\Delta TA_{2400}$, which upon wall cleaning sank to the sediment traps. This direct consequence of the abiotic precipitation increased ballasting material within the sediment trap and thus particle sinking velocities (Fig. 4c). Sinking velocities in the medium (100 – 250 µm) particle size class increased by 30 m d$^{-1}$ with every mol mol$^{-1}$ in $PIC_{ST}:POC_{ST}$. Although less discernible, particles in the smallest size range (25 – 100 µm) increased by 18.10 m d$^{-1}$ per increment of $PIC_{ST}:POC_{ST}$ (note marginally significance: p = 0.065, Table S4). While screen-captures from the FlowCam indicated, that larger particles were actually ballasted (distinct and dark particles in aggregates, Fig. 4d), solitary precipitates were also visible. It appears that both, ballasting of larger aggregates within the sediment trap and solitary precipitates increased the measured particle sinking velocities across all sizes (Fig. S5).

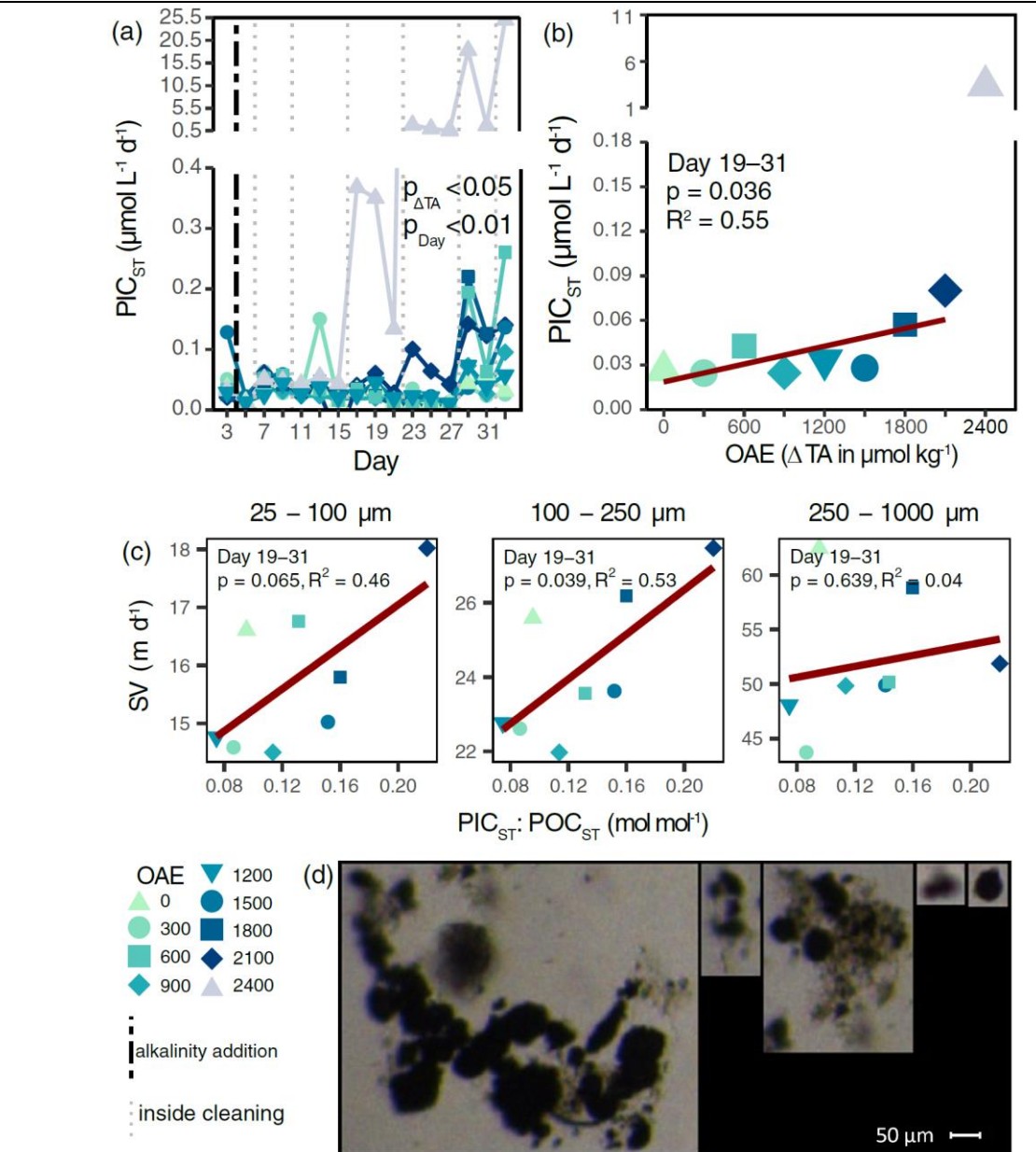

**Figure 4| Inorganic export flux and carbonate ballasting.** Inorganic carbon export and its response to OAE over time (a) and as averaged over the treatment period (b). p-value in (a) represents the output of linear mixed effects model (Table S3g). The vertical black line indicates the alkalinity addition and the grey lines the inside cleaning of the mesocosm walls. Average particle sinking velocities of different size classes and their response to carbonate ballasting during the phase of abiotic precipitation (c). Selection of FlowCam screen-captures of sediment trap material on Day 23 displaying carbonate ballasting of aggregates and individual carbonate precipitates (d).

## 4 Discussion

### 4.1 Particle export by the plankton community

Ocean alkalinity enhancement did not influence the biotically driven flux of major elements (C, N, P, Si), stoichiometric ratios or potential depth attenuation (particle sinking velocities and remineralization rates). These findings are in line with the lack of a detectable effect on plankton community composition or water column biogeochemistry during our experiment (Paul et al., 2024). Yet, the sporadically occurring blooms within some mesocosms ($\Delta TA_{600}$, $\Delta TA_{900}$, $\Delta TA_{1500}$, and $\Delta TA_{1800}$) were unexpected, considering the oligotrophic conditions within our mesocosms. In general, the community displayed a typical assemblage of open ocean gyres (Dai et al., 2023) and was dominated by picoeukaryotes (e.g. *Synechococcus*) and to lesser extent by diatoms (Xin et al., 2025). The bloom forming species was identified by Xin et al. (2025) as *Chrysochromulina spp.*, a non-calcifying haptophyte species with a cyanobacterial endosymbiont capable of nitrogen fixation (UCYN-A, Suzuki et al., 2021), allowing them to thrive in the otherwise nutrient-depleted mesocosm. Additionally, potential responses of calcifiers to OAE (due to an increase in $\Omega$ and pH) were either absent or not detected due to their low abundances (Fig. S8a-c), limiting conclusions of their influence on carbon export. This contrasted theoretical expectations (Jokiel, 2011; Monteiro et al., 2016; Bach, 2015) and we restrict this section to a general discussion of what might have caused the absence or disappearance of calcifying organisms during our experiment. Calcifying organisms are scarce in nutrient-poor seasons around the Canary Islands (Sprengel et al., 2002), underlining that a relevant seeding of mesocosms, similar to previous studies (Bach et al., 2017; Taucher et al., 2017; Bach et al., 2019c; Ortiz et al., 2022), might not have occurred. Additionally, Ortiz et al. (2022) only reported a significant contribution of calcifiers to the phytoplankton community upon repeated deep-water addition, implying that their potential competitive advantages under OAE (resource allocation towards growth and metabolic processes compared to calcification) could not be thoroughly expressed under the nutrient-deprived conditions of our study. In a recent alkalinity enhancement study, Gately et al. (2023) come to similar conclusions, claiming that nutrient concentrations and ratios, compared to alkalinity, may be a stronger driver governing phytoplankton community composition. Hence, the nutrient-deprived ecosystem, in suite with the deliberately small carbonate chemistry perturbations (only changing pH by up to 0.2 units) allowed us to assess this least impactful OAE scenario, not translating to detectable effects on calcifiers nor the remaining phytoplankton community (Xin et al., 2025) and the elemental fluxes they drive. This is consistent with previous ocean acidification studies, which found pronounced responses only at larger pH differences and during blooms (Stange et al., 2018; Taucher et al., 2017). Therein, particle fluxes only changed for a pH decrease of 0.5 units and upon adding nutrient-rich deep water to the oligotrophic system, which stimulated considerable primary production, in contrast to our study as inferred from unchanged water column chlorophyll *a* and POC concentrations.

Carbon-to-nitrogen ratios of exported matter remained unchanged under OAE and are in the same range as ratios found during previous mesocosm studies (Stange et al., 2018; Bach et al., 2019c; Baumann et al., 2021) and for natural un-alkalized export fluxes in the Canary Island region (Freudenthal et al., 2001). Our findings contrast Taucher et al. (2021), where ocean acidification altered carbon-to-nitrogen ratios due to varying heterotrophic respiration. While pH dependent microbial N-cycle

processes theoretically could affect these ratios (Fumasoli et al., 2017; Pommerening-Röser and Koops, 2005; Beman et al., 2011), we want to note, that observed optimal pH for these processes usually fall below the modest pH range implemented in this study. Instead, we attribute the stability of carbon-to-nitrogen ratios in our study to the nutrient-deprived conditions in combination with small pH changes (compared to Taucher et al. 2021), once again mirroring a lack of a detectable community effect under OAE.

While there is a chance that OAE effects might have emerged with longer experiment duration, added alkalinity for different application scenarios (e.g., in wake of ships) is expected to dilute on time scales considerably shorter than our study-length (Byrne et al., 1988). This gives further confidence that the particle fluxes, their properties and therefore the potential to sequester carbon via the biological pump would remain unimpeded in the oligotrophic open ocean under $CO_2$ pre-equilibrated OAE.

Other approaches, however, let $p$CO$_2$ equilibrate post alkalinity application and might be conducted in nutrient-rich ecosystems. Carbonate chemistry perturbations thus can be more severe during OAE and responses of plankton communities thereto might even be amplified during blooms, ultimately altering the biological carbon pump. For instance, non-equilibrated OAE could lead to $CO_2$ limitation in phytoplankton, which could hamper primary and subsequently biomass production (Bach et al., 2020; Zagarese et al., 2021; Riebesell et al., 1993) with the potential to decrease export production. Non-equilibrated alkalinity enhancement also entails a stronger pH change at which effects absent at milder perturbations might eventually emerge, analogous to several ocean acidification studies (Stange et al., 2018; Taucher et al., 2017). A lesson from strong pH perturbations under ocean acidification is that bacterial activity might increase (Piontek et al., 2010; Endres et al., 2014), such that a decrease under non-equilibrated OAE is conceivable. This would impair nutrient recycling and release in the surface ocean (Karthäuser et al., 2021; Cabanes et al., 2017), eventually shortening the bloom period (Böckmann et al., 2021; Sarthou et al., 2008; Cherabier and Ferrière, 2022) and consequently decrease the potential export production throughout a season (Iversen, 2023). On the other hand, copepods have been shown to increase their feeding rates in response to elevated pH (Li et al., 2008). Increased in- and egestion rates are thus conceivable under OAE, potentially leading to a larger amount of fast sinking fecal pellets (Stukel et al., 2011; Le Moigne et al., 2016), which eventually would decrease the depth attenuation of the export flux (Guidi et al., 2015).

While oligotrophic conditions in the Canary Island region are prevailing, eddy-induced nutrient upwelling is not uncommon (Arístegui et al., 1997; Basterretxea et al., 2002). Additionally, application of OAE is also considered in nutrient-rich coastal zones or upwelling areas, since it would require less technical efforts compared to open ocean applications, as represented by our study (Bach et al., 2019a). Thus, alkalinity enhanced water parcels may experience blooms. In parallel to several ocean acidification studies the blooming communities' sensitivity towards the concomitant carbonate chemistry perturbations can alter particle fluxes and emissions of climate relevant trace gases (Bach et al., 2016; Webb et al., 2016; Riebesell et al., 2017; Stange et al., 2018; Taucher et al., 2017). Thus, feedbacks of OAE on the biological carbon pump may be diverse, necessitating research on different application approaches and across nutrient regimes in further multi-trophic ecosystem studies.

## 4.2 Abiotic precipitation and alkalinity leakage

Particulate inorganic carbon fluxes increased with alkalinity enhancement; an effect that became more prominent with time. Yet, observed fluxes were apparently not driven from biotic production, given the low abundances of calcifying organisms during our mesocosm experiment (Xin et al., 2025; Fig. S8). Instead, the increased carbonate fluxes and the alkalinity loss observed in $\Delta TA_{2400}$ resulted from abiotic precipitation. This process was not unique to the highest treatment, as carbonate fluxes and water column ballasting ratios increased at lower treatment levels ($\Delta TA_{1800}$ and $\Delta TA_{2100}$) as well (Paul et al., 2024). Such abiotic precipitation under OAE has been observed previously (Moras et al., 2022; Griffioen, 2017; Subhas et al., 2022) and is known to occur for carbonate saturation states well below the threshold for spontaneous precipitation (Table S1: $\Omega_{Ar} = 10.01$ and $\Omega_{Ca} = 15.24$, compared to $\Omega_{Ar} = 12.5\text{-}13.5$, Morse and He, 1993). In fact, Wurgaft et al. (2021) noted carbonate precipitation at alkalinity concentrations close to our control treatment. The process can be initiated by the presence of suitable precipitation nuclei which has been noted in the field (Wurgaft et al., 2021) and during separate short-term incubations with mesocosm water during our campaign (Hartmann et al., 2023). Yet, it is important to note, that thresholds for spontaneous precipitation remain poorly constrained in natural ecosystems. This arises primarily from uncertainties regarding the influence of colloids and organic materials as precipitation nuclei, as well as diverse ranges of temperature and salinity, both affecting the kinetics and thermodynamics of secondary precipitation (Moras et al., 2022; Hartmann et al., 2023, 2013; Marion et al., 2009). However, artificial surfaces for precipitation were present through the mesocosm walls, which in combination with the perturbed carbonate chemistry may explain the observed precipitation at our higher alkalinity levels. Thus, a large portion of the formed PIC could be regarded an experimental artefact, which needs to be considered, when interpreting particle sinking velocities. The majority of precipitated carbonates slid down the mesocosm walls upon cleaning, so that most ballasting likely occurred through aggregation within the sediment trap where both PIC from the walls and POC from the water column accumulated. Given this spatial separation of inorganic and organic carbon flux production, it seems improbable that significant ballasting of aggregates and increase of sinking velocities occurred through scavenging of particles within the water column. However, being a strong predictor for particle sinking velocities (Honjo et al., 2008; Armstrong et al., 2001; Bach et al., 2016), carbonate ballasting yielded the prominent positive relation to particle sinking velocities seen in Fig. 4c, in contrast to OAE itself (Fig. 3b). The plankton community structure, likewise a strong predictor for particle sinking velocities (Bach et al., 2019c, 2016) did not display a response to OAE. Instead its natural variability among mesocosms likely obscured the detection of significant changes in sinking velocities with OAE. As such, alkalinity itself yielded weaker relations to particle sinking velocities, as compared to the experimental artefact of enhanced carbonate ballasting through abiotic precipitation. However, we still want to conclude that abiotic precipitation likely did not affect sinking velocities within the water column such that the increase over time could not have actually affected the potential depth attenuation of the biological pump during our experiment. Additionally, the magnitude of PIC formation in open ocean settings would likely be substantially lower than during our study, since suitable precipitation nuclei are mostly scarce.

Abiotic precipitation has also been noted for suspended particles (Ferderer et al., 2022; Hartmann et al. 2023). It is thus not only imaginable for coastal ecosystems with large surface areas (e.g., seagrass meadows), but also for the open ocean, given the periodically presence of precipitation nuclei through e.g., dust deposition. Although OAE should be maintained well outside a TA range where such precipitation might occur, it is worthwhile to consider its potential implications on particle fluxes should such a worst-case scenario arise. Carbonate ballasting from OAE-induced precipitates is likely to behave differently, as compared to e.g., coccolithophore-derived ballasting. Opposing to abiotic precipitates, biological carbonate-containing particles also include lighter organic matter, which exert less ballast on sinking aggregates. In addition, the abiotic PIC particles were considerably larger (50 μm) than e.g. coccolithophore-derived particles (Jordan, 2009; Rothwell, 2016). Larger particles have been shown to be scavenged and ballast sinking aggregates preferably (Puig et al., 2013; McCave, 1983). Abiotic precipitates are thus not only heavier at similar size, but potentially also larger, exerting more ballasting potential than many biologically derived carbonates and may substantially alter open ocean export dynamics under such adverse OAE scenarios. The onset of abiotic carbonate precipitation would also oppose the desired effects of ocean alkalinity enhancement. Given the supersaturation of surface waters with respect to calcium carbonate (Mucci, 1983), a significant export of the precipitated carbonate mineral is conceivable, transporting alkalinity to waters out of contact with the atmosphere. Such calcium carbonate precipitation releases $CO_2$ (Zeebe and Wolf-Gladrow, 2010), further decreasing OAE's efficacy to store atmospheric carbon. Additionally, abiotic carbonate precipitation may sustain itself to a point were alkalinity drops below initial background concentrations (Moras et al., 2022; Brečević and Nielsen, 1989; Hartmann et al., 2023), eventually turning alkalinity enhanced waters into a source, rather than a sink for $CO_2$. To conclude, our concerns of raising alkalinity to levels inducing precipitation extend beyond the $CO_2$ released upon carbonate precipitation but to potential adverse effects on the biological carbon pump as well.

## 5 Conclusions

Our findings suggest that particle fluxes by subtropical pelagic communities are insensitive to $CO_2$ pre-equilibrated OAE, leaving carbon sequestration via the biological pump unimpeded by this most optimistic approach. It will be paramount to test effects on the magnitude and quality of export fluxes across nutrient-richer ecosystems and for non-equilibrated OAE approaches. Undesired abiotic carbonate precipitation may induce alkalinity consumption and requires close monitoring. It may not only decrease OAE`s efficacy to store atmospheric carbon, but potentially also alters the biological pump`s capability to sequester atmospheric $CO_2$. Ultimately, our results are a promising starting point for further assessments on this carbon dioxide removal technology, prompting the necessity to holistically consider biotic and abiotic impacts, before drawing sound conclusions on the safe applicability of OAE.

**Acknowledgements**

The authors would like to thank the Oceanic Platform of the Canary Islands (PLOCAN) and its staff for the use of their facilities and for their help with the logistics and organisation of this experiment. We would also like to express our gratitude towards Andrea Ludwig, Jana Meyer, Jan Hennke, and Anton Theileis for logistical and technical support during and before our study.
Further, we are thankful for the help of the KOSMOS Scientific Diving and Maintenance Team, Michael Sswat, Carsten Spisla, Daniel Brüggemann, Silvan Urs Goldenberg, Joaquin Ortiz, Nicolás Sánchez. Additionally, we would like to thank Levka Hansen and Kerstin Nachtigall for laboratory support in Kiel and at last Anna Groen, Juliane Tammen and Julieta Schneider for further laboratory work and sample analysis, both on-site and in Kiel. We would also like to thank the Finnish marine research infrastructure (FINMARI) for providing relevant equipment to conduct our measurements.

**Data availability**

The datasets presented in this study can be found in an online repository. The name of the repository and accession numbers are: PANGEA, https://doi.org/10.1594/PANGAEA.967359

**Author contribution statement**

Study design and conceptualisation: PS, JT, SUG, MB, UR
Sampling and laboratory analysis: PS, AN, MV, KS
Data analysis and interpretation: PS, MB, SUG, JT, UR
Writing – original draft: PS
Writing – reviewing and editing: all authors

**Funding**

This study was funded by the OceanNETS project ("Ocean-based Negative Emissions Technologies – analysing the feasibility, risks and co-benefits of ocean-based negative emission technologies for stabilizing the climate", EU Horizon 2020 Research and Innovation Programme Grant Agreement No.: 869357), and the Helmholtz European Partnering project Ocean-CDR ("Ocean-based carbon dioxide removal strategies", Project No.: PIE-0021) with additional support from the AQUACOSM-plus project (EU H2020-INFRAIA Project No.: 871081, "AQUACOSM-plus: Network of Leading European AQUAtic
MesoCOSM Facilities Connecting Rivers, Lakes, Estuaries and Oceans in Europe and beyond").

**Competing interests**

The contact author has declared that none of the authors has any competing interests.

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
