# Peer review of "Particle fluxes by subtropical pelagic communities under ocean alkalinity enhancement"

_EGUsphere, 2023_

## Author Response (AR1)

**RC1**: 'Comment on egusphere-2023-2800', Anonymous Referee #1, 23 Jan 2024

- **AC2**: 'Reply on RC1', Philipp Suessle, 01 Mar 2024

Suessle et al. is a beautifully designed and conducted research on a particularly important topic – biological effect of Ocean Alkalinity Enhancement. The paper includes massive amount of novel data, and has some interesting conclusions. I recommend publishing the paper with some minor-moderate revisions. Below are my questions/concerns.

*Thank you for your insightful and constructive feedback on our manuscript. We appreciate the time and effort you have dedicated to carefully reviewing our work. Your comments provide valuable insights that will strengthen the quality of our research. We are committed to addressing each of your points in detail in the following.*

Comment 1: My main concern is regarding sinking velocity in Figure 3 and 4 – if PIC export and PIC/POC increases with dTA (Figure 4b and S6b), why wouldn't sinking velocity also increase with dTA (Figure 3b)? Can the authors elaborate on this topic a little bit in the manuscript?

*Response 1: Thank you for pointing this out. This is something we also expected previously. However, with the following we hope to clarify, why an increase of particle sinking velocities could not clearly be inferred from alkalinity concentrations.*

*Carbonate production and ballasting was directly increased by alkalinity enhancement as an effect of abiotic precipitation (Fig. 4b, Fig. S6b). Additionally, carbonate ballasting has been proven to be a strong predictor for particle sinking velocities (Honjo et al., 2008; Armstrong et al., 2001; Bach et al., 2016) Hence, the strong correlation observed between carbonate ballasting and particle sinking velocities demonstrated in Fig. 4c is not surprising. However, the plankton community structure has also been identified to be a strong predictor for particle sinking velocities (Bach et al., 2019, 2016). The natural variability of the plankton community during our experiment, rather than a clear treatment-driven driven response, likely obscured the detection of significant changes in particle sinking velocities with OAE. Therefore, increased carbonate ballasting and its relation as a more direct explanatory variable as compared to a more indirect explanatory variable, such as alkalinity, yielded stronger relationships with particle sinking velocities.*

*We hope by adding respective improvements to the discussion 4.2. (lines 406 – 411) will clarify this:*

*"However, being a strong predictor for particle sinking velocities (Honjo et al., 2008; Armstrong et al., 2001; Bach et al., 2016), carbonate ballasting yielded the prominent positive relation to particle sinking velocities seen in Fig. 4c, in contrast to OAE itself (Fig. 3b). The plankton community structure, likewise a strong predictor for particle sinking velocities (Bach et al., 2019, 2016) did not display a response to OAE. Instead its natural variability among mesocosms likely obscured the detection of significant changes in sinking velocities with OAE. As such, alkalinity itself yielded weaker relations to particle sinking velocities, as compared to the experimental artefact of enhanced carbonate ballasting through abiotic precipitation."*

Comment 2: What is the saturation state of the highest OAE level? How does it compare to previous studies that show the threshold saturation state when seawater starts to precipitate CaCO3? If this experiment is repeatable, could this be considered the maximum TA for OAE application?

*Response 2: Thank you for pointing this out. We hope to offer further insights in the following:*

*The carbonate saturation levels observed in the highest treatment level where precipitation was observed were $\Omega_{Ar} = 10.01$ and $\Omega_{Ca} = 15.24$. Experimental determinations indicate that carbonate precipitation from seawater, under the environmental conditions of our study (T = 22°C, PSU = 36), would only occur at $\Omega_{Ar} = 11.3$ and $\Omega_{Ca} = 18.8$ (Marion et al., 2009; Morse and He, 1993). While the carbonate saturation states during our study were considerably below these reported values, it is important to note that the thresholds for precipitation in natural ecosystems remain poorly constrained. This ambiguity arises primarily from uncertainties regarding the influence of colloids and organic materials as precipitation nuclei, as well as the diverse range of environmental conditions (T, PSU) that influence the kinetics and thermodynamics of secondary precipitation (Moras et al., 2022; Hartmann et al., 2023, 2013; Marion et al., 2009). Thus, direct comparisons of precipitation thresholds with studies conducted in other ecosystems are challenging. Nonetheless, we aim to provide contextual examples to assess whether the maximum alkalinity enhancement employed in our study could be considered a safe threshold for OAE application.*

*For instance, in Subhas et al. (2022), environmental conditions (T, PSU) closely resembled ours, yet carbonate precipitation in alkalinity-enhanced phytoplankton assemblages occurred only after prolonged sample storage (months as opposed to weeks in our study) and at lower carbonate saturation states ($\Omega_{Ar} \approx 6$ and $\Omega_{Ca} \approx 10$). In contrast, Gately et al. (2023) observed rapid carbonate precipitation (within 4 days) in mono-culture experiments with Chaetoceros spp. under similar environmental conditions to ours, but at a higher $\Omega_{Ca}$ value of 16.9. Moreover, during side experiments of a subsequent mesocosm campaign in Bergen, Norway (Suitner et al., in submission) alkalinity in experiments with natural plankton assemblages remained stable over 20 days, despite a carbonate saturation state ($\Omega_{Ar} \approx 12.0$) exceeding the critical $\Omega_{Ar}$ value (11.3) for pseudo-homogeneous precipitation (Marion et al., 2009). Although salinity was similar to our study, temperatures during the Suitner et al. experiments were considerably lower (12 – 15°C). These diverse findings underscore the highly variable nature of thresholds for carbonate precipitation and experiments determining the safe operating range for OAE in any given ecosystem and season should be prioritized. Hence, we want to note, that precipitation thresholds during our study may fluctuate throughout a season as environmental conditions change.*

*While most biological parameters remained unaffected during the event of precipitation, we contend that raising alkalinity to such concentrations may cause run-away precipitation, opposing desired effects of ocean alkalinity enhancement. This concern extends beyond the $CO_2$ released upon carbonate precipitation to potential effects on the biological carbon pump should precipitation persist over longer durations.*

*By extending our discussion in 4.2. (lines 395 – 399 and lines 433 – 435), we hope to have clarified, whether or not the highest alkalinity level could serve as a maximum TA concentration for the environmental boundary conditions given during our study.*

*Lines 395 – 399:*

*"Yet, it is important to note, that thresholds for spontaneous precipitation remain poorly constrained in natural ecosystems. This arises primarily from uncertainties regarding the influence of colloids and organic materials as precipitation nuclei, as well as diverse ranges of temperature and salinity, both affecting the kinetics and thermodynamics of secondary precipitation (Moras et al., 2022; Hartmann et al., 2023; Marion et al., 2009)."*

*Lines 433 – 435:*

*"To conclude, our concerns of raising alkalinity to levels inducing precipitation extend beyond the $CO_2$ released upon carbonate precipitation but to potential adverse effects on the biological carbon pump as well. "*

Comment 3: Figure 1b – why does regenerated nitrogen source only promote blooms in some mesocosms? Why is carbon biomass unaffected by the blooms?

*Response 3: Thanks for pointing this out. We want to clarify, that declaring the nitrogen sources as "regenerated" was misleading. With the following discussion we hope to resolve this error.*

*Considering the depletion of nutrients in all mesocosms, the emergence of significant phytoplankton blooms (measured as Chl a and POC concentrations) in the latter part of the experiment was unexpected and clearly unrelated to inorganic nutrient concentrations (N, P).*

*We want to highlight observations made by Xin et al. (in preparation) who identified potential species responsible for the observed blooms.*

*Xin et al. (in preparation) identified a non-calcifying and motile species of the haptophyte group, Chrysochromulina spp. as the main contributor to observed Chl a peaks in the blooming mesocosms. One potential species, Chrysochromulina polylepis, has been associated with high affinities for ammonium (Kaas et al., 1991). Additionally, Xin et al. (in preparation) suggests Chrysochromulina parkeae, a mixotrophic haptophyte with a cyanobacterial endosymbiont capable of nitrogen fixation (UCYN-A; Suzuki et al., 2021), allowing it to thrive in highly nutrient-depleted environments. The distribution of the UCYN-A nitrogenase gene sequence within our study region (Thompson et al., 2014) and the lower $^{15}N$ enrichment of particulate nitrogen in the blooming mesocosms (Paul et al. in preparation), support the idea of primary producers utilizing different nitrogen sources (e.g., $N_2$ fixation).*

*Why the emergence of Chrysochromulina spp. and blooms only occurred in some and not all mesocosms remains to be resolved. However, it is a common finding in mesocosm-based research that individual experimental ecosystem units may develop slightly differently from initial conditions (Stewart et al., 2013), meaning that communities enclosed may vary by chance (e.g., natural variability of enclosed water bodies) and not by treatment. Such treatment unrelated artefacts of enclosure have been noted in previous studies (Mine Berg et al., 1999; Domingues et al., 2023, and references therein). Possessing the capability of nitrogen fixation, Chrysochromulina spp. may have had a competitive advantage over other phytoplankton species, increasing its chance of taking into effect stronger so in some mesocosms (e.g., due to species specific predation relief), as compared to others.*

*Thus, we not only made according changes in lines 254 - 255 to clarify that blooming mesocosms consisted of Chrysochromulina spp., but also extended our discussion in 4.1. (line*

*322 – 327) to shed some more light on the general community composition and the occurrence of unexpected blooms:*

*"Yet, the sporadically occurring blooms within some mesocosms ($\Delta TA_{600}$, $\Delta TA_{900}$, $\Delta TA_{1500}$, and $\Delta TA_{1800}$) were unexpected, considering the oligotrophic conditions within our mesocosms. In general, the community displayed a typical assemblage of open ocean gyres (Dai et al., 2023) and was dominated by picoeukaryotes (e.g. Synechococcus) and to lesser extent by diatoms (Xin et al. in preparation). The bloom forming species was identified by Xin et al. (in preparation) as Chrysochromulina spp., a non-calcifying haptophyte species with a cyanobacterial endosymbiont capable of nitrogen fixation (UCYN-A, Suzuki et al., 2021), allowing them to thrive in the otherwise nutrient-depleted mesocosm."*

*Additionally, we want to point out, that phytoplankton biomass approximated from chlorophyll a and particulate organic carbon from within the water column ($POC_{WC}$) remained unaffected by alkalinity (lines 250 – 251):*

*"Phytoplankton biomass approximated by chlorophyll a and particulate organic carbon in the water column ($POC_{WC}$) did not change with respect to OAE (Fig. 1b,c, Table S4)."*

*But also want to mention, that $POC_{WC}$ did increase in response to the bloom, but simply to lesser extent, then chlorophyll a (lines 251 – 253):*

*"It low in the first half of the experiment (Day 3 – 21 Fig. S3), where after chlorophyll a and $POC_{WC}$ concentrations unexpectedly increased in some mesocosms ($\Delta TA_{600}$, $\Delta TA_{900}$, $\Delta TA_{1500}$, and $\Delta TA_{1800}$)."*

*We hope that by referring to the proxies of phytoplankton biomass (Chl a and $POC_{WC}$), the relation becomes clearer.*

Some minor comments:

Comment 4: Line 54, what do you mean by "CaCO3 precipitation is simplified"?

*Response 4: Lime-based ocean alkalinity enhancement increases the $CaCO_3$ saturation state by introducing an excess of $Ca_{2+}$ and $CO_3^{2-}$ ions into the seawater, while simultaneously raising pH levels. Both of these parameters have been established as robust predictors for the rate of calcification in marine phytoplankton (Bach, 2015; Monteiro et al., 2016, and references therein).*

*Calcification in marine organisms involves several energy-demanding steps, often regulated by electrochemical potentials across different cellular membranes, such as the plasma membrane or membranes of intracellular calcifying compartments (Monteiro et al., 2016). For instance, these energetic costs include the transport of $Ca^{2+}$ ions into the golgi-derived "coccolith vesicle" (CV), a process influenced by the saturation state ($\Omega = [Ca^{2+}]*[CO_3^{2-}]/K_{sp} > 1$, where $K_{sp}$ is the solubility constant). Further, alkalization within the CV, necessary for promoting $CaCO_3$ precipitation, is dependent on the electrochemical potential gradient of pH across the plasma membrane.*

*By mentioning the dependence of calcification on both the $CaCO_3$ saturation state and elevated pH in lines 53 – 56, we hope to clarify why calcifying organisms stand to benefit from ocean alkalinity enhancement:*

*"In turn, calcifying organisms may theoretically benefit from e.g. lime-based alkalinity enhancement, as the enhanced $CaCO_3$ saturation state (through excess of $Ca^{2+}$ and $CO_3^{2-}$ ions) and increased pH may facilitate $CaCO_3$ precipitation, allowing calcifiers to allocate more energy towards growth or other metabolic processes (Jokiel, 2011; Monteiro et al., 2016; Bach, 2015)."*

Comment 5: Line 89, suggest changing "side-effect" to "effect".

*Response 5: Changed as recommended.*

Comment 6: Line 95, I would suppose the cylinder is transparent – maybe adding "transparent" before cylindrical polyurethane foil.

*Response 6: Changed as recommended.*

Comment 7: Line 134, Paul et al., 2023 could not be found in the references. Are you referring to Paul et al. (in prep) as cited elsewhere in the manuscript?

*Response 7: Thank you for pointing out. Yes, we were referring to Paul et al. in preparation. Changed as recommended.*

Comment 8: Line 144-146, I had to refer to Boxhammer et al., 2016 to understand why FeCl3 and NaOH were added. I would suggest the authors add a simple sentence to explain – e.g., Adding FeCl3 and NaOH to increase flocculation and coagulation was shown to effectively promote the particle collection process by previous research etc.

*Response 8: Changed as recommended.*

Comment 9: Line 228-230, I don't understand the sentence starting with "OAE was employed as…". The authors should address what are "categorical fixed effect" and "random effect". I also couldn't find the reference Pinheiro et al., 2020 online.

*Response 9: We apologize for the misleading language used and want to resolve. Employed as continuous was meant to clarify that OAE is treated as a continuous fixed effect. We hope that by changes made in lines 235 – 236 the details of how statistics were conducted become clearer:*

*"OAE was set as a continuous fixed effect, Day as categorical fixed effect and Mesocosm as random effect (nlme package; Pinheiro et al., 2022)."*

*Additionally, we added the link to the R-package repository of the package from Pinheiro et al. (2022) and updated it to the newest version.*

*see: "Pinheiro, J., Bates, D., and R Core Team: nlme: Linear and Nonlinear Mixed Effects Models. R package version 3.1-160, https://CRAN.R-project.org/package=nlme, 2022."*

*References:*

Armstrong, R. A., Lee, C., Hedges, J. I., Honjo, S., and Wakeham, S. G.: A new, mechanistic model for organic carbon fluxes in the ocean based on the quantitative association of POC with ballast minerals, Deep Sea Research Part II: Topical Studies in Oceanography, 49, 219–236, https://doi.org/10.1016/S0967-0645(01)00101-1, 2001.

Bach, L. T.: Reconsidering the role of carbonate ion concentration in calcification by marine organisms, Biogeosciences, 12, 4939–4951, https://doi.org/10.5194/bg-12-4939-2015, 2015.

Bach, L. T., Boxhammer, T., Larsen, A., Hildebrandt, N., Schulz, K. G., and Riebesell, U.: Influence of plankton community structure on the sinking velocity of marine aggregates, Global Biogeochem Cycles, 30, 1145–1165, https://doi.org/10.1029/2019GB006256, 2016.

Bach, L. T., Stange, P., Taucher, J., Achterberg, E. P., Algueró-Muñiz, M., Horn, H., Esposito, M., and Riebesell, U.: The Influence of Plankton Community Structure on Sinking Velocity and Remineralization Rate of Marine Aggregates, Global Biogeochem Cycles, 33, 971–994, https://doi.org/10.1029/2019GB006256, 2019.

Dai, M., Luo, Y., Achterberg, E. P., Browning, T. J., Cai, Y., Cao, Z., Chai, F., Chen, B., Church, M. J., Ci, D., Du, C., Gao, K., Guo, X., Hu, Z., Kao, S., Laws, E. A., Lee, Z., Lin, H., Liu, Q., Liu, X., Luo, W., Meng, F., Shang, S., Shi, D., Saito, H., Song, L., Wan, X. S., Wang, Y., Wang, W., Wen, Z., Xiu, P., Zhang, J., Zhang, R., and Zhou, K.: Upper Ocean Biogeochemistry of the Oligotrophic North Pacific Subtropical Gyre: From Nutrient Sources to Carbon Export, Reviews of Geophysics, 61, https://doi.org/10.1029/2022RG000800, 2023.

Domingues, R. B., Mosley, B. A., Nogueira, P., Maia, I. B., and Barbosa, A. B.: Duration, but Not Bottle Volume, Affects Phytoplankton Community Structure and Growth Rates in Microcosm Experiments, Water (Basel), 15, 372, https://doi.org/10.3390/w15020372, 2023.

Gately, J. A., Kim, S. M., Jin, B., Brzezinski, M. A., and Iglesias-Rodriguez, M. D.: Coccolithophores and diatoms resilient to ocean alkalinity enhancement: A glimpse of hope?, Sci Adv, 9, https://doi.org/10.1126/sciadv.adg6066, 2023.

Hartmann, J., West, A. J., Renforth, P., Köhler, P., La Rocha, C. L., Wolf-Gladrow, D. A., Dürr, H. H., and Scheffran, J.: Enhanced chemical weathering as a geoengineering strategy to reduce atmospheric carbon dioxide, supply nutrients, and mitigate ocean acidification, Reviews of Geophysics, 51, 113–149, https://doi.org/10.1002/rog.20004, 2013.

Hartmann, J., Suitner, N., Lim, C., Schneider, J., Marín-Samper, L., Arístegui, J., Renforth, P., Taucher, J., and Riebesell, U.: Stability of alkalinity in ocean alkalinity enhancement (OAE) approaches – consequences for durability of CO2 storage, Biogeosciences, 20, 781–802, https://doi.org/10.5194/bg-20-781-2023, 2023.

Honjo, S., Manganini, S. J., Krishfield, R. A., and Francois, R.: Particulate organic carbon fluxes to the ocean interior and factors controlling the biological pump, Prog Oceanogr, 76, 217–285, https://doi.org/10.1016/j.pocean.2007.11.003, 2008.

Jokiel, P. L.: The reef coral two compartment proton flux model, J Exp Mar Biol Ecol, 409, 1–12, https://doi.org/10.1016/j.jembe.2011.10.008, 2011.

Marion, G. M., Millero, F. J., and Feistel, R.: Precipitation of solid phase calcium carbonates and their effect on application of seawater S-T-P models, Ocean Science, 5, 285–291, https://doi.org/10.5194/os-5-285-2009, 2009.

Mine Berg, G., Glibert, P. M., and Chen, C.-C.: Dimension effects of enclosures on ecological processes in pelagic systems, Limnol Oceanogr, 44, 1331–1340, https://doi.org/https://doi.org/10.4319/lo.1999.44.5.1331, 1999.

Monteiro, F., Bach, L. T., Brownlee, C., Bown, P., E, R. R., Poulton, A. J., Tyrrell, T., Beaufort, L., Dutkiewicz, S., Gibbs, S., Gutowska, M. A., Lee, R., Riebesell, U., Young, J., and Ridgwell, A.: Why marine phytoplankton calcify, Sci Adv, 2, e1501822, https://doi.org/10.1126/sciadv.1501822, 2016.

Moras, C. A., Bach, L. T., Cyronak, T., Joannes-Boyau, R., and Schulz, K. G.: Ocean alkalinity enhancement - avoiding runaway CaCO3 precipitation during quick and hydrated lime dissolution, Biogeosciences, 19, 3537–3557, https://doi.org/10.5194/bg-19-3537-2022, 2022.

Morse, J. W. and He, S.: Influences of T, S and PCO2 on the pseudo-homogeneous precipitation of CaCO3 from seawater, Mar Chem, 41, 291–297, https://doi.org/10.1016/0304-4203(93)90261-L, 1993.

Pinheiro, J., Bates, D., and R Core Team: nlme: Linear and Nonlinear Mixed Effects Models. R package version 3.1-160, https://CRAN.R-project.org/package=nlme, 2022.

Stewart, R. I. A., Dossena, M., Bohan, D. A., Jeppesen, E., Kordas, R. L., Ledger, M. E., Meerhoff, M., Moss, B., Mulder, C., Shurin, J. B., Suttle, B., Thompson, R., Trimmer, M., and Woodward, G.: Chapter Two - Mesocosm Experiments as a Tool for Ecological Climate-Change Research, in: Advances in Ecological Research, vol. 48, edited by: Woodward, G. and O'Gorman, E. J., Academic Press, 71–181, https://doi.org/https://doi.org/10.1016/B978-0-12-417199-2.00002-1, 2013.

Subhas, A. V, Marx, L., Reynolds, S., Flohr, A., Mawji, E. W., Brown, P. J., and Cael, B. B.: Microbial ecosystem responses to alkalinity enhancement in the North Atlantic Subtropical Gyre, Frontiers in Climate, 4, https://doi.org/10.3389/fclim.2022.784997, 2022.

Suitner, N., Faucher, G., Lim, C., Schneider, J., Moras, C. A., Riebesell, U., and Hartmann, J.: Ocean alkalinity enhancement approaches and the predictability of runaway precipitation processes - Results of an experimental study to determine critical alkalinity ranges for safe and sustainable application scenarios, EGUsphere, 2023, 1–35, https://doi.org/10.5194/egusphere-2023-2611, 2023.

Suzuki, S., Kawachi, M., Tsukakoshi, C., Nakamura, A., Hagino, K., Inouye, I., and Ishida, K.: Unstable Relationship Between Braarudosphaera bigelowii (= Chrysochromulina parkeae) and Its Nitrogen-Fixing Endosymbiont, Front Plant Sci, 12, https://doi.org/10.3389/fpls.2021.749895, 2021.

Thompson, A., Carter, B. J., Turk-Kubo, K., Malfatti, F., Azam, F., and Zehr, J. P.: Genetic diversity of the unicellular nitrogen-fixing cyanobacteria UCYN-A and its prymnesiophyte host, Environ Microbiol, 16, 3238–3249, https://doi.org/10.1111/1462-2920.12490, 2014.

**RC2**: , Anonymous Referee #2, 29 Jan 2024

- **AC1**: , Philipp Suessle, 01 Mar 2024

'Comment on egusphere-2023-2800', Anonymous Referee #2, 29 Jan 2024

Suessle et al. provide a comprehensively conducted research on the application of OAE using mesocosms. The resulting dataset contains a multitude of parameters which support the conclusions made in this manuscript. The key finding of this study is, that alkalinity enhancement via the presented equilibrated approach and thus keeping seawater carbonate chemistry largely undisturbed does not translate to a change in particle properties. As such, this study provides I) evidence for OAE as a reasonably safe CDR technique in terms of biotic consequences and II) further evidence of potential abiotic precipitation, thus informing potential thresholds for efficient OAE application at scale.

*We would like to thank you for your insightful and constructive feedback on our manuscript. Your dedication to reviewing our work is truly appreciated. We believe that by addressing your insights the quality of our manuscript will be enhanced significantly and are fully committed to addressing each of your points in detail in our responses below.*

Comment 1: My main concern is regarding the measured biological parameters in this study. Here, only calcifying organisms were discussed and appear to have declined quite rapidly throughout the study period. It would be good to discuss other members of the community as well. This would be very useful in such a nutrient deprived system, where water column POC did not appear to change over the course of the incubation period. Can the authors elaborate a bit further on this?

*Response 1: Thank you for raising this concern. We have indeed given it considerable thought during the compilation of our manuscript. We believe that discussing other aspects of the community is warranted to gain a comprehensive understanding of why alkalinity enhancement did not influence water column biogeochemistry or particle fluxes and properties.*

*In the process of manuscript preparation, we prioritized avoiding overlap with the community-centered narrative being developed in the manuscript by Xin et al. This decision was made to maximize synergy from our experiment while ensuring that our primary focus on water column biogeochemistry and particle fluxes remained clear. It's worth noting that the majority of the community-centered narrative in Xin et al.'s manuscript also conveys a "no effect" story regarding alkalinity.*

*We chose to primarily discuss calcifying organisms for several reasons. Calcifying organisms and their potential responses to lime-based OAE have been and continue to be key concerns in alkalinity enhancement research (Bach et al., 2019a; Gately et al., 2023; Renforth and Henderson, 2017). Additionally, differences in the population sizes of calcifying phytoplankton species like Emiliania huxleyi have been linked not only to emissions of climate-relevant trace gases (Webb et al., 2016; Riebesell et al., 2017), but also to the magnitude and efficiency of particle fluxes reaching the deep ocean (Bach et al., 2016), lending their discussion, despite their rapid decline, special importance in the context of our study.*

*Nevertheless, we extended our discussion in 4.1. to also include a community description in order to fill in the reader with the necessary background information (lines 322 – 327), but prefer to keep the major discussion of OAE effects on community parameters to calcifying species (lines 327 – 341), especially in the light of a consistent "no effect" story across the prepared manuscripts (Xin et al., Paul et al., in preparation).*

*Lines 322 – 327:*

*"Yet, the sporadically occurring blooms within some mesocosms ($\Delta TA_{600}$, $\Delta TA_{900}$, $\Delta TA_{1500}$, and $\Delta TA_{1800}$) were unexpected, considering the oligotrophic conditions within our mesocosms. In general, the community displayed a typical assemblage of open ocean gyres (Dai et al., 2023) and was dominated by picoeukaryotes (e.g. Synechococcus) and to lesser extent by diatoms (Xin et al. in preparation). The bloom forming species was identified by Xin et al. (in preparation) as Chrysochromulina spp., a non-calcifying haptophyte species with a cyanobacterial endosymbiont capable of nitrogen fixation (UCYN-A, Suzuki et al., 2021), allowing them to thrive in the otherwise nutrient-depleted mesocosm."*

*Lines 327 – 341:*

*"Additionally, potential responses of calcifiers to OAE (due to an increase in $\Omega$ and pH) were either absent or not detected due to their low abundances (Fig. S8a-c), limiting conclusions of their influence on carbon export. This contrasted theoretical expectations (Jokiel, 2011; Monteiro et al., 2016; Bach, 2015) and we restrict this section to a general discussion of what might have caused the absence or disappearance of calcifying organisms during our experiment. Calcifying organisms are scarce in nutrient-poor seasons around the Canary Islands (Sprengel et al., 2002), underlining that a relevant seeding of mesocosms, similar to previous studies (Bach et al., 2017; Taucher et al., 2017; Ortiz et al., 2022; Bach et al., 2019c), might not have occurred. Additionally, Ortiz et al. (2022) only reported a significant contribution of calcifiers to the phytoplankton community upon repeated deep-water addition, implying that their potential competitive advantages under OAE (resource allocation towards growth and metabolic processes compared to calcification) could not be thoroughly expressed under the nutrient-deprived conditions of our study. In a recent alkalinity enhancement study, Gately et al. (2023) come to similar conclusions, claiming that nutrient concentrations and ratios, compared to alkalinity, may be a stronger driver governing phytoplankton community composition. Hence, the nutrient-deprived ecosystem, in suite with the deliberately small carbonate chemistry perturbations (only changing pH by up to 0.2 units) allowed us to assess this least impactful OAE scenario, not translating to detectable effects on calcifiers nor the remaining phytoplankton community (Xin et al. in preparation) and the elemental fluxes they drive."*

Comment 2: Likewise, the mention of potential blooming events in some of the mesocosms was related to regenerated nitrogen sources. However, this is difficult to justify without further insight into non-calcifying members of the community, both auto- and heterotrophic. Can the authors shed some light on this?

*Response 2: Thanks for pointing this out. We hope to clarify upon answering in the following:*

*Considering the depletion of nutrients in all mesocosms, the emergence of significant phytoplankton blooms (measured as chlorophyll a and POC concentrations) in the latter part of the experiment was unexpected and clearly unrelated to inorganic nutrient concentrations (N, P).*

*We want to highlight observations from Xin et al. (in preparation) who identified potential species responsible for the observed blooms.*

*Xin et al. (in preparation) identified a non-calcifying and motile species of the haptophyte group, Chrysochromulina spp. as the main contributor to observed Chl a peaks in the blooming mesocosms. One potential species, Chrysochromulina polylepis, has been associated with high affinities for ammonium (Kaas et al., 1991). Additionally, Xin et al. (in preparation) suggests Chrysochromulina parkeae, a mixotrophic haptophyte with a cyanobacterial endosymbiont capable of nitrogen fixation (UCYN-A; Suzuki et al., 2021), allowing it to thrive in highly nutrient-depleted environments. The distribution of the UCYN-A nitrogenase gene sequence within our study region (Thompson et al., 2014) and the lower $^{15}N$ enrichment of particulate nitrogen in the blooming mesocosms (Paul et al. in preparation), support the idea of primary producers utilizing different nitrogen sources (e.g., $N_2$ fixation).*

*To date, why the emergence of Chrysochromulina spp. and blooms only occurred in some and not all mesocosms remains to be resolved. However, we want to hint towards a long-standing paradigm in mesocosm related research. Experimental ecosystem units may display a "random-walk" from initial conditions (Stewart et al., 2013), meaning that communities enclosed may vary by chance and not by treatment. Such treatment unrelated artefacts of enclosure have been noted in previous studies (Mine Berg et al., 1999; Domingues et al., 2023, and references therein). Possessing the capability of heterotrophy, Chrysochromulina spp. may have had a competitive advantage over other phytoplankton species, increasing its chance of taking into effect stronger so in some mesocosms (e.g., due to species specific predation relief), as compared to others.*

*We hope, that by adding lines 322 – 327 (see above), we could address you concerns and can help potential readers to judge our conclusions with this necessary background information. We believe however, that keeping the discussion on the community aspect as little as possible would be beneficial for carrying across the main message of our manuscript, especially in the light of such random bloom developments.*

Comment 3: Lastly, were their potential bottle effects observed? Nitrogen levels are really low, as expected, but there was seemingly a drop in phosphate (FigS2C) over the first two weeks of the incubation period.

*Response 3: Initial phosphate concentrations during our study decreased despite the absence of nitrate and water column Chl a and POC concentrations remaining stable.*

*Recent alkalinity enhancement studies have also reported changes in phosphate concentrations (Gately et al., 2023; Subhas et al., 2022). Gately et al. (2023) offer a potential mechanism: $Ca^{2+}$ ions at pH values above 8.5 have been documented to remove phosphate in Ca-phosphate precipitates (Zhang et al., 2020), likely explaining the observed phosphate losses in their moderate (pH = 8.44) and high (pH = 8.87) alkalinity treatments. However, since our highest alkalinity treatment did not surpass pH = 8.3, the onset of carbonate precipitation was only observed after phosphate concentrations stabilized (Day 13/15 Fig S2 vs. Day 17/19 Fig. 4) and the decrease in phosphate did not correlate with alkalinity (Fig S2 and Paul et al. in preparation Fig. 4), we do not believe this mechanism accounts for the initial phosphate decrease during our experiment.*

*Instead, Subhas et al. (2022) proposed an alkalinity-independent explanation: a decoupling of nitrate and phosphate related to mixotrophy in the North Atlantic Oligotrophic Gyre*

*(Hartmann et al., 2012), where bacterial over algal uptake of phosphate may dominate (Zubkov et al., 2007). Shifts from autotrophic towards heterotrophic biomass have also been observed as enclosure effects during bottle incubations (Calvo-Díaz et al., 2011). Coupled with the absence of an alkalinity effect in our study, this explanation may offer a more plausible account for the observed initial phosphate decrease.*

*Another plausible explanation could involve the uptake of phosphate by picoeukaryotes, which are known to possess highly efficient phosphate transporters even at low substrate levels (Lomas et al., 2014). Upon filling the mesocosms, there is a truncation of the community, characterized by the exclusion of larger grazers. Additionally, the plankton community experiences higher irradiance within the mesocosms compared to the source location, attributable to the reduced vertical mixing range. Consequently, and consistent with observations from prior mesocosm studies conducted on Gran Canaria, an initial shift in the community towards a predominance of picoeukaryotes may potentially explain the observed phosphate consumption.*

*However, we cannot fully resolve the decline of phosphate, but believe, that it does not affect the general conclusions with respect to alkalinity enhancement made in this study.*

*As the primary aim of our study was to investigate the effects of ocean alkalinity enhancement on the biological carbon pump, and any potential phosphate uptake likely occurred only in the first half of the experiment, we did not address this potential enclosure effect in our manuscript.*

Despite these concerns, the manuscript is a strong piece of work and addresses known issues in relation to abiotic precipitation, which is discussed in depth by the authors. Some of the aforementioned concerns might be answered upon publication of Xin et al. and Paul et al., as it seems that these studies complement each other?

*We have taken note of your aforementioned concerns and hope to clarify them in our responses to previous comments. Additionally, the forthcoming publications by Xin et al. and Paul et al. will provide further clarity and context to our findings, as these studies complement our own.*

Some minor comments:

Comment 4: Line 54-55: Here the authors state "… allowing them to allocate more energy towards growth…". However, the calcifying organisms decline over the incubation period (Fig S8). Can you elaborate on this in the discussion?

*Response 4: Thanks for pointing this out. With the improvements implemented in lines 327 – 341 (see above), we hope to have sufficiently clarified your concerns.*

Comment 5: Line 77-79: A change in particulate elemental ratios has not been observed in the present study, changing heterotrophic processes could have an influence on this. Can the authors elaborate on this further in the discussion?

*Response 5: Thanks for pointing this out. With an expansion of the discussion in lines 346 – 354, we hope to shed some light on why elemental ratios did not change under ocean alkalinity enhancement:*

*"Carbon-to-nitrogen ratios of exported matter remained unchanged under OAE and are in the same range as ratios found during previous mesocosm studies (Stange et al., 2018; Bach et al., 2019c; Baumann et al., 2021) and for natural un-alkalized export fluxes in the Canary Island region (Freudenthal et al., 2001). Our findings contrast Taucher et al. (2021), where ocean acidification altered carbon-to-nitrogen ratios due to varying heterotrophic respiration. While pH dependent microbial N-cycle processes theoretically could affect these ratios (Fumasoli et al., 2017; Pommerening-Röser and Koops, 2005; Beman et al., 2011), we want to note, that observed optimal pH for these processes usually falls below the modest pH range implemented in this study. Instead, we attribute the stability of carbon-to-nitrogen ratios in our study to the nutrient-deprived conditions in combination with small pH changes (compared to Taucher et al. 2021), once again mirroring a lack of a detectable community effect under OAE."*

Comment 6: Experimental setup:

- Were the mesocosms deployed in the harbour for easier access and are the environmental settings similar to place where the water was sourced from?
- It would be good to have some reference in regards to environmental conditions, i.e. temperature, salinity, turbidity

*Response 6: Yes, the mesocosms were deployed in the harbor of Taliarte (27° 59' 24" N, 15° 22' 8" W; east coast of Gran Canaria, Spain) for easier access during sampling. Additional CTD casts (CTD 60M, Sea and Sun Technology) were conducted prior to treatment addition and were carried out every two days throughout the experiment. Temperature, salinity, and photosynthetically active radiation (PAR) profiles measured before and after treatment addition reflected typical conditions for the Canary Island region during the given season (Barton et al., 1998), similar to the environmental conditions from where the water for mesocosm filling was sourced.*

*The reduced vertical mixing range in the mesocosms results in significantly higher irradiance on the enclosed communities compared to natural settings due to the enclosing nature of a mesocosm system. Further, it is important to acknowledge that the light regime (PAR) within our mesocosms may have been further influenced compared to the surrounding water due to the presence of transparent polyurethane foil. However, we made efforts to minimize potential shading from the pier or other structures in the harbor by employing a mooring and pulley system, ensuring that the mesocosms remained 2 meters away from the pier during non-sampling times. We have addressed these efforts to maintain environmental conditions as close to natural as possible in lines 96 – 98 (pulley system).*

*Additionally, we aimed to clarify that CTD measurements conducted prior to alkalinity addition ensured that the environmental conditions within the mesocosms were representative of those expected for the surface mixed layer of the given oceanographic setting and season. Therefore, the following changes were made in lines 103 – 106:*

*"CTD casts (CTD60M, Sea & Sun Technology GmbH, Trappenkamp, Germany) for temperature, salinity, pH, turbidity, oxygen ($O_2$) and photosynthetically active radiation prior to treatment addition assured, that the environmental conditions of water pumped into the mesocosms were as close as possible to those expected for the surface mixed layer of the given oceanographic setting and season (Barton et al., 1998)."*

*CTD data will be made available upon publication on a data repository (PANGEA).*

Comment 7: Line 98-99: Why from 2 to 10 meters? Was that within the mixed layer?

*Response 7: Various scenarios for ocean alkalinity enhancement involve increasing alkalinity near the sea surface or within the surface mixed layer (Bach et al., 2019a; Köhler et al., 2013; Rau et al., 2013; Feng et al., 2017). While elevated surface alkalinity can gradually penetrate into the ocean's interior through mixing, the time frame for this process can vary from weeks to several years, depending on the oceanographic setting (Liu et al., 2019 and references therein).*

*Although investigating the effects of alkalinity enhancement on sub-surface communities upon mixing is essential for future studies, it is crucial to recognize that the most pronounced effects may occur in surface ocean communities, precisely where alkalinity concentrations are highest upon addition. Therefore, prioritizing the testing of biological and biogeochemical effects of ocean alkalinity enhancement on surface ocean communities is imperative.*

*In the Canary Island region, during the period from September to October, the depth of the surface mixed layer can extend to approximately 50 meters (Barton et al., 1998), thereby ensuring that the composition of our mesocosm seed community mirrors that of surface ocean communities. Moreover, the water depth at the collection site was approximately 20 meters, allowing for the extraction of water from depths ranging between 2 and 10 meters. This strategy served as a practical compromise, enabling us to capture surface ocean communities while circumventing the collection of sediment or benthic organisms. By including the respective reference and noting that the sampling depth was a compromise between entrapping surface ocean communities whilst avoiding benthic impact, we aim to address this concern in lines 98 – 101:*

*"The mesocosms were filled with seawater from ~100 m outside the harbor (water depth at collection site ~ 20 m). Using a peristaltic pump, water from 2 – 10 m depth (within the surface mixed layer, Barton et al., 1998, but avoiding benthic impact) was filtered (< 3 mm) and gathered in an intermediate storage volume, from where each mesocosm simultaneously received ~8 $m^3$ in total."*

Comment 8: Line 109: What is this custom-built device, and how was the even spread of solution confirmed/traced?

*Response 8: The custom-built device utilized to uniformly distribute the alkalinity treatment across the diameter and water column of the mesocosm is described in Riebesell et al., 2013, Figure 7 and referred to as the "spider." This apparatus consists of a polyoxymethylene body, surrounded by a 5kg stainless steel stand, through which treatment solutions can be pumped via a garden hose. The solution is then dispersed into elastic acrylic branches of varying lengths, ensuring even distribution of the treatment solution across the diameter of the mesocosm. By moving the "spider" steadily up and down throughout the water column, treatment can be evenly distributed throughout the entire mesocosm. We have included the reference Riebesell et al., 2013 in line 114.*

*Elevating alkalinity is accompanied by a pH increase from the baseline (Zeebe and Wolf-Gladrow, 2010). CTD casts for every mesocosm on the day following the alkalinity addition revealed a homogenous increase of pH across the entire water column of the mesocosms, proportional to the amount of alkalinity added. CTD data will be made available upon publication on a data repository (PANGEA).*

Comment 9: Line 124-126: Which methods were employed to undertake these measurements and are they being presented in Xin et al. or Paul et al.?

*Response 9: We highlighted the parameters in line 124 – 126 (now lines 129 – 131) not only to illustrate the breadth of biological and biogeochemical analyses conducted to comprehensively characterize and monitor the responses to alkalinity enhancement in our mesocosm system, but also because these parameters were utilized to assess any potential effects on the magnitude and efficiency of the biological carbon pump. However, upon data exploration, analyses of these parameters did not yield meaningful insights regarding the influence of alkalinity enhancement on the biological carbon pump. Consequently, detailed data for most of these parameters was not presented or described methodologically.*

*Comprehensive descriptions of primary productivity and prokaryotic heterotrophic production measurements can be found in Marín-Samper et al. (2023, preprint), while details regarding photosynthetic pigment analysis are provided in Xin et al. (in preparation) and Paul et al. (in preparation). Furthermore, phyto- and microzooplankton analysis details are available in Xin et al. (in preparation).*

*However, we do acknowledge, that data concerning phyto- and microzooplankton analysis was not only used to infer on the low and rapidly decreasing abundances of calcifying organisms, but also presented in Fig. S8. As mentioned, detailed methods can be found in Xin et al. in preparation, but a brief description thereof was now added to this manuscript (lines 216 - 217) to ensure completeness in terms of methods, results and respective interpretation.*

*"Phyto- and microzooplankton abundances were determined by microscope from bulk water samples according to Utermöhl (1958) after fixating in basics Lugols solution and letting samples settle in a chamber for 24 h."*

*Additionally, we do recognize, that comparisons done with ocean acidifications influence on primary production (discussion 4.1.) might suggest that it was done likewise in the present study. However, we want to clarify, that this was done via the extrapolation of Chl a and water column POC concentrations as direct consequences and thus proxies of increased primary production. We hope to clarify this by mentioning the respective inference in lines 343 – 354:*

*"Therein, particle fluxes only changed for a pH decrease of 0.5 units and upon adding nutrient-rich deep water to the oligotrophic system, which stimulated considerableprimary production, in contrast to our study as inferred from unchanged water column chlorophyll a and POC concentrations."*

Comment 10: Line 127: Some of these parameters are discussed in this manuscript and would be useful, maybe in a supplementary table?

*Response 10: Thanks for pointing this out. Indeed, CTD data was used at several occasions during the experiment and for compiling this manuscript. CTD data will be made available upon publication on a data repository (PANGEA).*

Figure 1: TA seemed to have increased across all treatments and over time. Was the water collected for sampling replenished or did this increase could have come from increasing evaporation within the mesocosms?

The water collected for sampling was not replenished during the experiment. The slight increase of TA over the course of the experiment can indeed be related to evaporation. Air temperatures and irradiation during our experiment were high and are not uncommon for the Canary Island region. A slight increase of PSU (derived by CTD casts) from day 1 (PSU = 36.5) until the end of the experiment (PSU = 37.35) supports this.

However, it appears that evaporation occurred uniformly across all mesocosms, thereby maintaining the applied gradient of TA across them. Moreover, the extent of TA increase due to evaporation is expected to be minimal compared to the applied TA concentrations, particularly given the absence of any discernible biological or biogeochemical effects during our study, even at elevated TA concentrations. CTD data will be made available upon publication on a data repository (PANGEA).

Comment 11: Line 315-317: Considering the quite drastic decline in calcifying organisms (or in the case of coccolithophores basically not detectable), can responses really be expected?

Response 11: We hope by responding to your comment on lines 54 – 55, we could clarify this concern as well.

References:

Bach, L. T.: Reconsidering the role of carbonate ion concentration in calcification by marine organisms, Biogeosciences, 12, 4939–4951, https://doi.org/10.5194/bg-12-4939-2015, 2015.

Bach, L. T., Boxhammer, T., Larsen, A., Hildebrandt, N., Schulz, K. G., and Riebesell, U.: Influence of plankton community structure on the sinking velocity of marine aggregates, Global Biogeochem Cycles, 30, 1145–1165, https://doi.org/10.1029/2019GB006256, 2016.

Bach, L. T., Alvarez-Fernandez, S., Hornick, T., Stuhr, A., and Riebesell, U.: Simulated ocean acidification reveals winners and losers in coastal phytoplankton, PLoS One, 12, 1–22, https://doi.org/10.1371/journal.pone.0188198, 2017.

Bach, L. T., Gill, S. J., Rickaby, R. E. M., Gore, S., and Renforth, P.: CO2 Removal With Enhanced Weathering and Ocean Alkalinity Enhancement, Frontiers in Climate, 1:7, https://doi.org/10.3389/fclim.2019.00007, 2019a.

Bach, L. T., Hernández-Hernández, N., Taucher, J., Spisla, C., Sforna, C., Riebesell, U., and Arístegui, J.: Effects of Elevated CO2 on a Natural Diatom Community in the Subtropical NE Atlantic, Front Mar Sci, 6:75, 320, https://doi.org/10.3389/fmars.2019.00075, 2019b.

Bach, L. T., Stange, P., Taucher, J., Achterberg, E. P., Algueró-Muñiz, M., Horn, H., Esposito, M., and Riebesell, U.: The Influence of Plankton Community Structure on Sinking Velocity and Remineralization Rate of Marine Aggregates, Global Biogeochem Cycles, 33, 971–994, https://doi.org/10.1029/2019GB006256, 2019c.

Barton, E. D., Arístegui, J., Tett, P., Cantón, M., García-Braun, J., Hernández-León, S., Nykjaer, L., Almeida, C., Almunia, J., Ballesteros, S., Basterretxea, G., Escánez, J., García-Weill, L., Hernández-Guerra, A., López-Laatzen, F., Molina, R., Montero, M. F., Navarro-Pérez, E., Rodríguez, J. M., van Lenning, K., Vélez, H., and Wild, K.: The transition zone of

the Canary Current upwelling region, Prog Oceanogr, 41, 455–504, https://doi.org/https://doi.org/10.1016/S0079-6611(98)00023-8, 1998.

Baumann, M., Taucher, J., Paul, A. J., Heinemann, M., Vanharanta, M., Bach, L. T., Spilling, K., Ortiz, J., Arístegui, J., Hernández-Hernández, N., Baños, I., and Riebesell, U.: Effect of Intensity and Mode of Artificial Upwelling on Particle Flux and Carbon Export, Front Mar Sci, 8, https://doi.org/10.3389/fmars.2021.742142, 2021.

Beman, J. M., Chow, C.-E., King, A. L., Feng, Y., Fuhrman, J. A., Andersson, A., Bates, N. R., Popp, B. N., and Hutchins, D. A.: Global declines in oceanic nitrification rates as a consequence of ocean acidification, Proceedings of the National Academy of Sciences, 108, 208–213, https://doi.org/10.1073/pnas.1011053108, 2011.

Calvo-Díaz, A., Díaz-Pérez, L., Suárez, L. Á., Morán, X. A. G., Teira, E., and Marañón, E.: Decrease in the autotrophic-to-heterotrophic biomass ratio of picoplankton in oligotrophic marine waters due to bottle enclosure, Appl Environ Microbiol, 77, 5739–5746, 2011.

Dai, M., Luo, Y., Achterberg, E. P., Browning, T. J., Cai, Y., Cao, Z., Chai, F., Chen, B., Church, M. J., Ci, D., Du, C., Gao, K., Guo, X., Hu, Z., Kao, S., Laws, E. A., Lee, Z., Lin, H., Liu, Q., Liu, X., Luo, W., Meng, F., Shang, S., Shi, D., Saito, H., Song, L., Wan, X. S., Wang, Y., Wang, W., Wen, Z., Xiu, P., Zhang, J., Zhang, R., and Zhou, K.: Upper Ocean Biogeochemistry of the Oligotrophic North Pacific Subtropical Gyre: From Nutrient Sources to Carbon Export, Reviews of Geophysics, 61, https://doi.org/10.1029/2022RG000800, 2023.

Domingues, R. B., Mosley, B. A., Nogueira, P., Maia, I. B., and Barbosa, A. B.: Duration, but Not Bottle Volume, Affects Phytoplankton Community Structure and Growth Rates in Microcosm Experiments, Water (Basel), 15, 372, https://doi.org/10.3390/w15020372, 2023.

Feng, E. Y., Koeve, W., Keller, D. P., and Oschlies, A.: Model-Based Assessment of the CO2 Sequestration Potential of Coastal Ocean Alkalinization, Earths Future, 5, 1252–1266, https://doi.org/10.1002/2017EF000659, 2017.

Freudenthal, T., Neuer, S., Meggers, H., Davenport, R., and Wefer, G.: Influence of lateral particle advection and organic matter degradation on sediment accumulation and stable nitrogen isotope ratios along a productivity gradient in the Canary Islands region, Mar Geol, 177, 93–109, https://doi.org/10.1016/S0025-3227(01)00126-8, 2001.

Fumasoli, A., Bürgmann, H., Weissbrodt, D. G., Wells, G. F., Beck, K., Mohn, J., Morgenroth, E., and Udert, K. M.: Growth of Nitrosococcus-Related Ammonia Oxidizing Bacteria Coincides with Extremely Low pH Values in Wastewater with High Ammonia Content, Environ Sci Technol, 51, 6857–6866, https://doi.org/10.1021/acs.est.7b00392, 2017.

Gately, J. A., Kim, S. M., Jin, B., Brzezinski, M. A., and Iglesias-Rodriguez, M. D.: Coccolithophores and diatoms resilient to ocean alkalinity enhancement: A glimpse of hope?, Sci Adv, 9, https://doi.org/10.1126/sciadv.adg6066, 2023.

Hartmann, M., Grob, C., Tarran, G. A., Martin, A. P., Burkill, P. H., Scanlan, D. J., and Zubkov, M. V: Mixotrophic basis of Atlantic oligotrophic ecosystems, Proceedings of the National Academy of Sciences, 109, 5756–5760, 2012.

Jokiel, P. L.: The reef coral two compartment proton flux model, J Exp Mar Biol Ecol, 409, 1–12, https://doi.org/10.1016/j.jembe.2011.10.008, 2011.

Köhler, P., Abrams, J. F., Völker, C., Hauck, J., and Wolf-Gladrow, D. A.: Geoengineering impact of open ocean dissolution of olivine on atmospheric $CO_2$, surface ocean pH and marine biology, Environmental Research Letters, 8, 014009, https://doi.org/10.1088/1748-9326/8/1/014009, 2013.

Liu, X., Dunne, J. P., Stock, C. A., Harrison, M. J., Adcroft, A., and Resplandy, L.: Simulating Water Residence Time in the Coastal Ocean: A Global Perspective, Geophys Res Lett, 46, 13910–13919, https://doi.org/10.1029/2019GL085097, 2019.

Lomas, M. W., Bonachela, J. A., Levin, S. A., and Martiny, A. C.: Impact of ocean phytoplankton diversity on phosphate uptake, Proceedings of the National Academy of Sciences, 111, 17540–17545, https://doi.org/10.1073/pnas.1420760111, 2014.

Marín-Samper, L., Arístegui, J., Hernández-Hernández, N., Ortiz, J., Archer, S. D., Ludwig, A., and Riebesell, U.: Assessing the impact of CO2 equilibrated ocean alkalinity enhancement on microbial metabolic rates in an oligotrophic system, EGUsphere, 2023, 1–29, https://doi.org/10.5194/egusphere-2023-2409, 2023.

Mine Berg, G., Glibert, P. M., and Chen, C.-C.: Dimension effects of enclosures on ecological processes in pelagic systems, Limnol Oceanogr, 44, 1331–1340, https://doi.org/https://doi.org/10.4319/lo.1999.44.5.1331, 1999.

Monteiro, F., Bach, L. T., Brownlee, C., Bown, P., E, R. R., Poulton, A. J., Tyrrell, T., Beaufort, L., Dutkiewicz, S., Gibbs, S., Gutowska, M. A., Lee, R., Riebesell, U., Young, J., and Ridgwell, A.: Why marine phytoplankton calcify, Sci Adv, 2, e1501822, https://doi.org/10.1126/sciadv.1501822, 2016.

Ortiz, J., Arístegui, J., Taucher, J., and Riebesell, U.: Artificial Upwelling in Singular and Recurring Mode: Consequences for Net Community Production and Metabolic Balance, Front Mar Sci, 8, https://doi.org/10.3389/fmars.2021.743105, 2022.

Pommerening-Röser, A. and Koops, H.-P.: Environmental pH as an important factor for the distribution of urease positive ammonia-oxidizing bacteria, Microbiol Res, 160, 27–35, https://doi.org/10.1016/j.micres.2004.09.006, 2005.

Rau, G. H., Carroll, S. A., Bourcier, W. L., Singleton, M. J., Smith, M. M., and Aines, R. D.: Direct electrolytic dissolution of silicate minerals for air $CO_2$ mitigation and carbon-negative $H_2$ production, Proceedings of the National Academy of Sciences, 110, 10095–10100, https://doi.org/10.1073/pnas.1222358110, 2013.

Renforth, P. and Henderson, G.: Assessing ocean alkalinity for carbon sequestration, Reviews of Geophysics, 55, 636–674, https://doi.org/10.1002/2016RG000533, 2017.

Riebesell, U., Czerny, J., von Bröckel, K., Boxhammer, T., Büdenbender, J., Deckelnick, M., Fischer, M., Hoffmann, D., Krug, S. A., Lentz, U., Ludwig, A., Muche, R., and Schulz, K. G.: Technical Note: A mobile sea-going mesocosm system – new opportunities for ocean change research, Biogeosciences, 10, 1835–1847, https://doi.org/10.5194/bg-10-1835-2013, 2013.

Riebesell, U., Bach, L. T., Bellerby, R. G. J., Monsalve, J., Boxhammer, T., Czerny, J., Larsen, A., Ludwig, A., and Schulz, K. G.: Competitive fitness of a predominant pelagic calcifier impaired by ocean acidification, Nat Geosci, 10, 19–23, https://doi.org/10.1038/ngeo2854, 2017.

Sprengel, C., Baumann, K.-H., Henderiks, J., Henrich, R., and Neuer, S.: Modern coccolithophore and carbonate sedimentation along a productivity gradient in the Canary Islands region, Deep Sea Research Part II: Topical Studies in Oceanography, 49, 3577–3598, https://doi.org/10.1016/S0967-0645(02)00099-1, 2002.

Stange, P., Taucher, J., Bach, L. T., Algueró-Muñiz, M., Horn, H. G., Krebs, L., Boxhammer, T., Nauendorf, A. K., and Riebesell, U.: Ocean Acidification-Induced Restructuring of the Plankton Food Web Can Influence the Degradation of Sinking Particles, Front Mar Sci, 5, https://doi.org/10.3389/fmars.2018.00140, 2018.

Stewart, R. I. A., Dossena, M., Bohan, D. A., Jeppesen, E., Kordas, R. L., Ledger, M. E., Meerhoff, M., Moss, B., Mulder, C., Shurin, J. B., Suttle, B., Thompson, R., Trimmer, M., and Woodward, G.: Chapter Two - Mesocosm Experiments as a Tool for Ecological Climate-Change Research, in: Advances in Ecological Research, vol. 48, edited by: Woodward, G. and O'Gorman, E. J., Academic Press, 71–181, https://doi.org/https://doi.org/10.1016/B978-0-12-417199-2.00002-1, 2013.

Subhas, A. V, Marx, L., Reynolds, S., Flohr, A., Mawji, E. W., Brown, P. J., and Cael, B. B.: Microbial ecosystem responses to alkalinity enhancement in the North Atlantic Subtropical Gyre, Frontiers in Climate, 4, https://doi.org/10.3389/fclim.2022.784997, 2022.

Suzuki, S., Kawachi, M., Tsukakoshi, C., Nakamura, A., Hagino, K., Inouye, I., and Ishida, K.: Unstable Relationship Between Braarudosphaera bigelowii (= Chrysochromulina parkeae) and Its Nitrogen-Fixing Endosymbiont, Front Plant Sci, 12, https://doi.org/10.3389/fpls.2021.749895, 2021.

Taucher, J., Bach, L. T., Boxhammer, T., Nauendorf, A., Consortium, T. G. C. K., Achterberg, E. P., Algueró-Muñiz, M., Arístegui, J., Czerny, J., Esposito, M., Guan, W., Haunost, M., Horn, H. G., Ludwig, A., Meyer, J., Spisla, C., Sswat, M., Stange, P., and Riebesell, U.: Influence of Ocean Acidification and Deep Water Upwelling on Oligotrophic Plankton Communities in the Subtropical North Atlantic, Front Mar Sci, 4, https://doi.org/10.3389/fmars.2017.00085, 2017.

Taucher, J., Boxhammer, T., Bach, L. T., Paul, A. J., Schartau, M., Stange, P., and Riebesell, U.: Changing carbon-to-nitrogen ratios of organic-matter export under ocean acidification, Nat Clim Chang, 11, 52–57, https://doi.org/10.1038/s41558-020-00915-5, 2021.

Thompson, A., Carter, B. J., Turk-Kubo, K., Malfatti, F., Azam, F., and Zehr, J. P.: Genetic diversity of the unicellular nitrogen-fixing cyanobacteria UCYN-A and its prymnesiophyte host, Environ Microbiol, 16, 3238–3249, https://doi.org/10.1111/1462-2920.12490, 2014.

Utermöhl, H.: Zur Vervollkommnung der quantitativen Phytoplankton Methodik, Schweizerbart Science Publishers, Stuttgart, Germany, 1958.

Webb, A. L., Malin, G., Hopkins, F. E., Ho, K. L., Riebesell, U., Schulz, K., Larsen, A., and Liss, P.: Ocean acidification affects production of DMS and DMSP measured in a mesocosm

study and cultures of Emiliania huxleyi and a mesocosm study:, Environ. Chem, 13, 314–329, https://doi.org/10.1071/EN14268, 2016.

Zeebe, R. E. and Wolf-Gladrow, D. A.: CO2 in seawater: Equilibrium, kinetics, isotopes, edited by: Zeebe, R. E. and Wolf-Gladrow, D. A., Elsevier, Amsterdam; New York, 346 pp., 2010.

Zhang, H., Elskens, M., Chen, G., Snoeck, C., and Chou, L.: Influence of seawater ions on phosphate adsorption at the surface of hydrous ferric oxide (HFO), Science of The Total Environment, 721, 137826, 2020.

Zubkov, M. V, Mary, I., Woodward, E. M. S., Warwick, P. E., Fuchs, B. M., Scanlan, D. J., and Burkill, P. H.: Microbial control of phosphate in the nutrient-depleted North Atlantic subtropical gyre, Environ Microbiol, 9, 2079–2089, 2007.